# ITERCOMP: ITERATIVE COMPOSITION-AWARE FEEDBACK LEARNING FROM MODEL GALLERY FOR TEXT-TO-IMAGE GENERATION

**Xinchen Zhang**[1*]   **Ling Yang**[2*]   **Guohao Li**[5]   **Yaqi Cai**[4]   **Jiake Xie**[3]   **Yong Tang**[3]
**Yujiu Yang**[1†]   **Mengdi Wang**[6]   **Bin Cui**[2†]
[1]Tsinghua University   [2]Peking University   [3]LibAI Lab   [4]USTC
[5]University of Oxford   [6]Princeton University
https://github.com/YangLing0818/IterComp

## ABSTRACT

Advanced diffusion models like Stable Diffusion 3, Omost, and FLUX have made notable strides in compositional text-to-image generation. However, these methods typically exhibit distinct strengths for compositional generation, with some excelling in handling attribute binding and others in spatial relationships. This disparity highlights the need for an approach that can leverage the complementary strengths of various models to comprehensively improve the composition capability. To this end, we introduce IterComp, a novel framework that aggregates composition-aware model preferences from multiple models and employs an iterative feedback learning approach to enhance compositional generation. Specifically, we curate a gallery of six powerful open-source diffusion models and evaluate their three key compositional metrics: attribute binding, spatial relationships, and non-spatial relationships. Based on these metrics, we develop a composition-aware model preference dataset comprising numerous image-rank pairs to train composition-aware reward models. Then, we propose an iterative feedback learning method to enhance compositionality in a closed-loop manner, enabling the progressive self-refinement of both the base diffusion model and reward models over multiple iterations. Detailed theoretical proof demonstrates the effectiveness of this method. Extensive experiments demonstrate our significant superiority over previous methods, particularly in multi-category object composition and complex semantic alignment. IterComp opens new research avenues in reward feedback learning for diffusion models and compositional generation.

## 1 INTRODUCTION

The rapid advancement of diffusion models (Sohl-Dickstein et al., 2015; Ho et al., 2020; Song et al., 2020; Peebles & Xie, 2023) has recently brought unprecedented progress to the field of text-to-image generation, with powerful models like DALL-E 3 (Betker et al., 2023), Stable Diffusion 3, (Esser et al., 2024) and FLUX (BlackForest, 2024) demonstrating remarkable capabilities in generating aesthetic and diverse images. However, these models often struggle to follow complex prompts to achieve precise compositional generation (Omost-Team, 2024; Yang et al., 2024b; Zhang et al., 2024b), which requires the model to possess robust, comprehensive capabilities in various aspects, such as attribute binding, spatial relationships, and non-spatial relationships (Huang et al., 2023).

To enhance compositional generation, some works introduce additional conditions such as layouts/boxes (Li et al., 2023; Zhou et al., 2024; Wang et al., 2024a; Zhang et al., 2024b). InstanceDiffusion (Wang et al., 2024a) controls the generation process using layouts, masks, or other conditions through trainable instance masked attention layers. Although these layout-based methods demonstrate strong spatial awareness, they struggle with image realism, especially in generating non-spatial relationships and preserving aesthetic quality (Zhang et al., 2024b). Another potential solution lever-

---

[*]Contributed equally. Contact: yangling0818@163.com
[†]Corresponding authors.

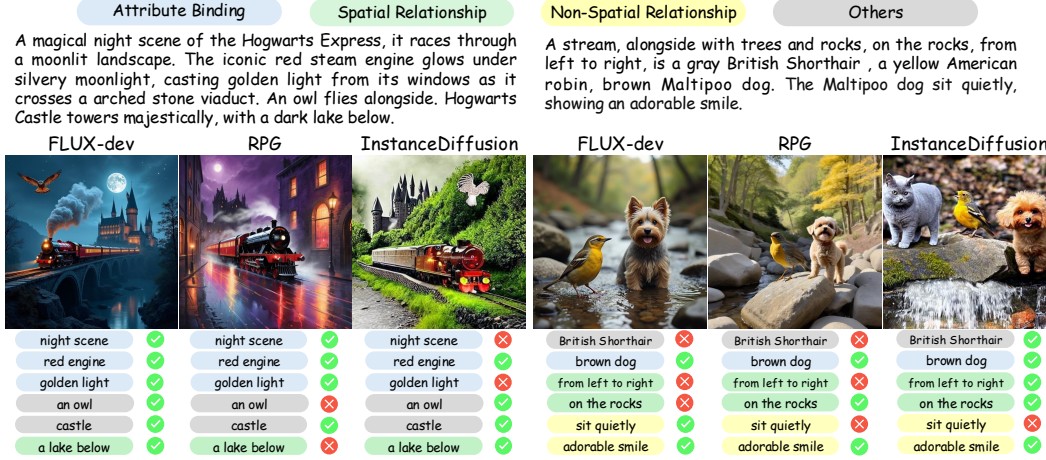

Figure 1: **Motivation of IterComp.** We select three types of compositional generation methods. The results show that different models exhibit distinct strengths across various aspects of compositional generation. fig. 3 further demonstrated these distinct strengths quantitatively.

ages the impressive reasoning abilities of Large Language Models (LLMs) to decompose complex generation tasks into simpler subtasks (Yang et al., 2024b; Omost-Team, 2024; Wang et al., 2024b). RPG (Yang et al., 2024b) employs MLLMs as the global planner to transform the process of generating complex images into multiple simpler generation tasks within subregions. However, it requires designing complex prompts for LLMs, and it is challenging to achieve precise generation results due to their intricate outputs (Yang et al., 2024b).

We conducted extensive experiments to explore the unique strengths of different models in compositional generation. As shown in the left example in fig. 1, text-to-image model FLUX (BlackForest, 2024) demonstrates impressive performance in attribute binding and aesthetic quality due to its advanced training techniques and model architecture. In contrast, layout-to-image model InstanceDiffusion (Wang et al., 2024a) struggles to capture fine-grained visual details, such as 'night scene' or 'golden light.' In the right example of fig. 1, where the text prompt involves complex spatial relationships between multiple objects, FLUX (BlackForest, 2024) exhibits limitations in spatial awareness. In contrast, InstanceDiffusion (Wang et al., 2024a) excels in handling spatial relationships through layout guidance. This demonstrates that different models exhibit distinct strengths across various aspects of compositional generation. Moreover, fig. 3 further demonstrated these distinct strengths quantitatively. Naturally, a pertinent question arises: *Is there a method capable of excelling in all aspects of compositional generation?*

In order to enable the diffusion model to improve compositional generation comprehensively, we present a new framework, ***IterComp***, which collects *composition-aware model preferences* from various models, and then employs a novel yet simple *iterative feedback learning* framework to achieve comprehensive improvements in compositional generation. Firstly, we select six open-sourced models excelling in different aspects of compositionality to form our model gallery. We focus on three essential compositional metrics: attribute binding, spatial relationships, and non-spatial relationships to curate a new composition-aware model preference dataset, which consists of a large number of image-rank pairs. Next, to comprehensively capture diverse composition-aware model preferences, we train reward models to provide fine-grained compositional guidance during the finetuning of the base diffusion model. Finally, given that compositional generation is difficult to optimize, we propose iterative feedback learning. This approach enhances compositionality in a closed-loop manner, allowing for the progressive self-refinement of both the base diffusion model and reward models in multiple iterations. We theoretically and experimentally demonstrate the effectiveness of our method and its significant improvement in compositional generation.

Our contributions are summarized as follows:

- We propose the first iterative composition-aware reward-controlled framework *IterComp*, to comprehensively enhance the compositionality of the base diffusion model.

- We curate a model gallery and develop a high-quality composition-aware model preference dataset comprising numerous image-rank pairs.

- We utilize a new *iterative feedback learning* framework to progressively enhance both the reward models and the base diffusion model.

- Extensive qualitative and quantitative comparisons with previous SOTA methods demonstrate the superior compositional generation capabilities of our approach.

## 2 RELATED WORK

**Compositional Text-to-Image Generation**    Compositional text-to-image generation is a complex and challenging task that requires a model with comprehensive capabilities, including the understanding of complex prompts and spatial awareness (Yang et al., 2024b; Zhang et al., 2024b). Some methods enhance prompt comprehension by using more powerful text encoders or architectures (Esser et al., 2024; Betker et al., 2023; Hu et al., 2024; Dai et al., 2023). Stable Diffusion 3 (Esser et al., 2024) utilizes three different-sized text encoders to enhance prompt comprehension. DALL-E 3 (Betker et al., 2023) enhances the understanding of rich textual details by expanding image captions through recaptioning. However, compositional capability such as spatial awareness remains a limitation of these models (Li et al., 2023; Chen et al., 2024a). Other methods attempt to enhance spatial awareness by the control of additional conditions (e.g., layouts) (Yang et al., 2023; Dahary et al., 2024). BoxDiff (Xie et al., 2023) and LMD (Lian et al., 2023b) guide the generated objects to strictly adhere to the layout by designing energy functions based on cross-attention maps. ControlNet (Zhang et al., 2023) and T2I-Adapter (Mou et al., 2024) specify high-level image features to control semantic structures. Although these methods enhance spatial awareness, they often compromise image realism (Zhang et al., 2024b). Additionally, some approaches leverage the powerful reasoning capabilities of LLMs to assist in the generation process (Yang et al., 2024b; Omost-Team, 2024; Wang et al., 2024b). RPG (Yang et al., 2024b) employs MLLM to decompose complex compositional generation tasks into simpler subtasks. However, these methods require designing complex prompts as inputs to the LLM, and the diffusion model struggles to produce precise results due to the LLM's intricate outputs (Yang et al., 2024b). In contrast, our method extracts these preferences from different models in model gallery and trains composition-aware reward models to refine the base diffusion model iteratively, achieving robust compositionality across multiple aspects.

**Diffusion Model Alignment**    Building on the success of reinforcement learning from human feedback (RLHF) in Large Language Models (LLMs) (Ouyang et al., 2022; Bai et al., 2022), numerous methods in diffusion models have attempted to use similar approaches for model alignment (Lee et al., 2023; Fan et al., 2024; Sun et al., 2023). Some methods use a pretrained reward model or train a new one to guide the generation process(Zhang et al., 2024a; Black et al., 2023; Deng et al., 2024; Clark et al., 2023; Prabhudesai et al., 2023). For instance, ImageReward (Xu et al., 2024) manually annotated a large dataset of human-preferred images and trained a reward model to assess the alignment between images and human preferences. Reward Feedback Learning (ReFL) is proposed for tuning diffusion models with the ImageReward model. RAHF (Liang et al., 2024a) is trained on RichHF-18K, a high-quality dataset rich in human feedback, and is capable of predicting the unreasonable parts in generated images. Some methods bypass the training of a reward model and directly finetune diffusion models on human preference datasets (Yang et al., 2024a; Liang et al., 2024b; Yang et al., 2024c). Diffusion-DPO (Wallace et al., 2024) reformulates Direct Preference Optimization (DPO) to account for a diffusion model's notion of likelihood, utilizing the evidence lower bound to derive a differentiable objective. The potential for alignment in diffusion models goes beyond this. We iteratively align the base model with composition-aware model preferences from the model gallery, effectively enhancing its performance on compositional generation.

## 3 METHOD

In this section, we present our method, IterComp, which collects composition-aware model preferences from the model gallery and utilizes iterative feedback learning to enhance the comprehensive capability of the base diffusion model in compositional generation. An overview of IterComp is illustrated in fig. 2. In section 3.1, we introduce the method for collecting the composition-aware model preference dataset from the model gallery. In section 3.2, we describe the training process for the composition-aware reward models and multi-reward feedback learning. In section 3.2, we

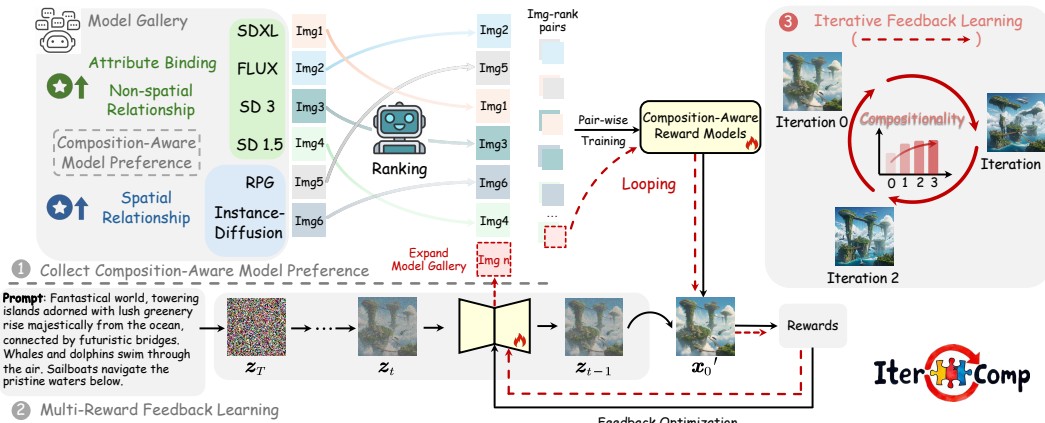

Figure 2: Overview of IterComp. We collect composition-aware model preferences from multiple models and employ an iterative feedback learning approach to enable the progressive self-refinement of both the base diffusion model and reward models.

propose the iterative feedback learning framework to enable the self-refinement of both the base diffusion model and reward models, progressively enhancing compositional generation.

### 3.1 COLLECTING HUMAN PREFERENCES OF COMPOSITIONALITY

**Compositional Metric and Model Gallery** We focus on three key aspects of compositionality: attribute binding, spatial relationships, and non-spatial relationships (Huang et al., 2023), to collect composition-aware model preferences. We initially select six open-sourced models excel in different aspects of compositional generation as our model gallery: FLUX-dev (BlackForest, 2024), Stable Diffusion 3 (Esser et al., 2024), SDXL (Podell et al., 2023), Stable Diffusion 1.5 (Rombach et al., 2022), RPG (Yang et al., 2024b), and InstanceDiffusion (Wang et al., 2024a).

**Human Ranking on Attribute Binding** For attribute binding, we randomly select 500 prompts from each of the following categories: color, shape, and texture in the T2I-CompBench (Huang et al., 2023), resulting in a total of 1,500 prompts. Three professional experts ranked the images generated by the six models for each prompt, and their rankings were weighted to determine the final result. The primary criterion is whether the attributes mentioned in the prompt were accurately reflected in the generated images, especially the correct representation and binding of attributes to the corresponding objects.

**Human Ranking on Complex Relationships** For spatial and non-spatial relationships, we select 1,000 prompts for each category from the T2I-CompBench (Huang et al., 2023) and apply the same manual annotation method to obtain the rankings. For spatial relationships, the primary ranking criterion is whether the objects are correctly generated and whether their spatial positioning matches the prompt. For non-spatial relationships, the focus is on whether the objects display natural and realistic actions.

**Analysis of Composition-aware Model Preference Dataset** For each prompt, we obtain 6 images and $\binom{6}{2} = 15$ image-rank pairs. As shown in table 1, in total, we collected a dataset with 22,500 image-rank pairs for model preference in attribute binding, 15,000 for spatial relationships, and 15,000 for non-spatial relationships. We visualize the proportion of generated images ranked first for each model in fig. 3. The results demonstrate that different models exhibit distinct strengths across various aspects of compositional generation, and this dataset effectively captures a diverse range of composition-aware model preferences.

### 3.2 COMPOSITION-AWARE MULTI-REWARD FEEDBACK LEARNING

**Composition-aware Reward Model Training** To achieve comprehensive improvements in compositional generation, we utilize three types of composition-aware datasets described in section 3.1,

Table 1: Statistics on the composition-aware model preference dataset. The dataset consists of 3,500 text prompts, 27,500 images, and 52,500 image-rank pairs.

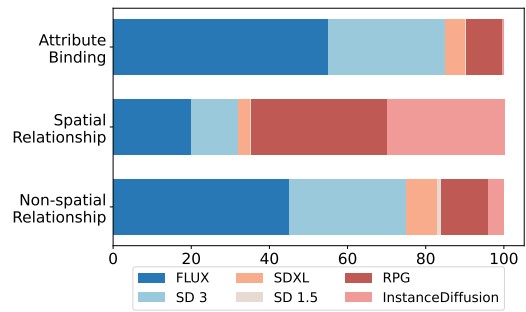

Figure 3: The proportion of each model ranked first.

|  | Counts | | |
| --- | --- | --- | --- |
| Category | Texts | Images | Image-rank pairs |
| Attribute Binding | 1,500 | 9,000 | 22,500 |
| Spatial Relationship | 1,000 | 6,000 | 15,000 |
| Non-spatial Relationship | 1,000 | 6,000 | 15,000 |
| Total | 3,500 | 21,000 | 52,500 |

decomposing compositionality into three subtasks and training a specific reward model for each. Specifically, the reward model $\mathcal{R}_{\theta_i}(c, x_0)$ is trained using the input format $x_0^w \succ x_0^l \mid c$, where $x_0^w$ and $x_0^l$ denoting the "winning" and "losing" images, $c$ denoting the text prompt. We select two images corresponding to the same prompt from the composition-aware model preference datasets to form an input image-rank pair, and trained the reward model using the following loss function:

$$\mathcal{L}(\theta_i) = -\mathbb{E}_{(c, x_0^w, x_0^l) \sim \mathcal{D}_i} \left[ \log \left( \sigma \left( \mathcal{R}_{\theta_i} \left( c, x_0^w \right) - \mathcal{R}_{\theta_i} \left( c, x_0^l \right) \right) \right) \right] \tag{1}$$

where $\mathcal{D}$ denotes the composition-aware model preference dataset, $\sigma(\cdot)$ is the sigmoid function.

The three composition-aware reward models apply BLIP (Li et al., 2022; Xu et al., 2024) as feature extractors. We combine the extracted image and text features with cross attention mechanism, and use a learnable MLP to generate a score scalar for preference comparison.

**Multi-Reward Feedback Learning**   Due to the multi-step denoising process in diffusion models, yielding likelihoods for their generations is impossible, making the RLHF approach used in language models unsuitable for diffusion models. Some existing methods (Xu et al., 2024; Zhang et al., 2024a) finetune diffusion models directly by treating the scores of the reward model as the human preference loss. To optimize the base diffusion model using multiple composition-aware reward models, we design the loss function as follows:

$$\mathcal{L}(\theta) = \lambda \mathbb{E}_{c_j \sim \mathcal{C}} \sum_i \left( \phi \left( \mathcal{R}_i \left( c_j, p_\theta \left( c_j \right) \right) \right) \right) \tag{2}$$

where $\mathcal{C} = \{c_1, c_2, \ldots, c_n\}$ denotes the prompt set, $p_\theta(c)$ denotes the generate image of diffusion model with parameter $\theta$ under the condition of prompt $c$. We calculate the loss for each reward model $\mathcal{R}_i(\cdot)$ and sum them to obtain the multi-reward feedback loss.

### 3.3   ITERATIVE OPTIMIZATION OF COMPOSITION-AWARE FEEDBACK LEARNING

Compositional generation is challenging to optimize due to its inherent complexity and multifaceted nature, requiring both our reward models and base diffusion model to excel in aspects such as complex text comprehension and the generation of complex relationships. To ensure more thorough optimization, we propose an iterative feedback learning framework that progressively refines both the reward models and the base diffusion model over multiple iterations.

At the $(k+1)$-th iteration of the optimization described in section 3.2, we denote the reward models and the base diffusion model from the previous iteration as $\mathcal{R}^k(\cdot)$ and $p_\theta^k(\cdot)$, respectively. For each prompt $c$ in the datasets $\mathcal{D}^k$, we sample an image $x_0^* = p_\theta^k(c)$ and expand the composition-aware model preference dataset $\mathcal{D}^k$ with the sampled image. The image rankings for each prompt are updated using the trained reward model $\mathcal{R}_\theta^k(\cdot)$, while preserving the relative ranks of the initial six images. Following this process, we update the composition-aware model preference dataset to a more comprehensive version, denoted as $\mathcal{D}^{k+1}$. Using this dataset, we finetune both the reward models and the base diffusion model to get $\mathcal{R}^{k+1}(\cdot)$ and $p_\theta^{k+1}(\cdot)$. The detailed process of iterative feedback learning can be found in algorithm 1.

**Effectiveness of Iterative Feedback Learning**   Through this iterative feedback learning framework, the reward models become more effective at understanding complex compositional prompts,

---

**Algorithm 1** Iterative Composition-aware Feedback Learning

---

**Dataset:** Composition-aware model preference dataset $\mathcal{D}_0 = \{((\boldsymbol{c}_1, \boldsymbol{x}_0^w, \boldsymbol{x}_0^l), \ldots, (\boldsymbol{c}_n, \boldsymbol{x}_0^w, \boldsymbol{x}_0^l)\}$
Prompt set $\mathcal{C} = \{\boldsymbol{c}_1, \boldsymbol{c}_2, \ldots, \boldsymbol{c}_n\}$
**Input:** Base model with pretrained parameters $p_\theta$, reward model $\mathcal{R}$, reward-to-loss map function $\phi$, reward re-weight scale $\lambda$, iterative optimization iterations $iter$
**Initialization:** Number of noise scheduler time steps $T$, time step range for finetuning $[T_1, T_2]$

1: **for** $k = 0, \ldots, iter$ **do**
2:     **for** $(\boldsymbol{c}_i, \boldsymbol{x}_0^w, \boldsymbol{x}_0^l) \in \mathcal{D}_k$ **do**
3:         $\mathcal{L} \leftarrow \log\left(\sigma\left(\mathcal{R}_{\theta_i}^k\left(\boldsymbol{c}, \boldsymbol{x}_0^w\right) - \mathcal{R}_{\theta_i}^k\left(\boldsymbol{c}, \boldsymbol{x}_0^l\right)\right)\right)$   // Reward model loss
4:         $\mathcal{R}_{\theta_{i+1}}^k \leftarrow \mathcal{R}_{\theta_i}^k(\boldsymbol{c}_i, \boldsymbol{x}_0^w, \boldsymbol{x}_0^l)$   // Update the reward models
5:     **end for**   // Get $\mathcal{R}^{k+1}$ after training
6:     **for** $\boldsymbol{c}_i \in \mathcal{C}$ **do**
7:         $t \leftarrow rand(T_1, T_2)$   // Pick a random timestep $t \in [T_1, T_2]$
8:         $\boldsymbol{z}_T \sim \mathcal{N}(\boldsymbol{0}, \mathbf{I})$
9:         **for** $j = T, \ldots, t+1$ **do**
10:            **no grad:** $\boldsymbol{z}_{j-1} \leftarrow p_{\theta_i}^k(\boldsymbol{z}_j)$
11:         **end for**
12:         **with grad:** $\boldsymbol{z}_{t-1} \leftarrow p_{\theta_i}^k(\boldsymbol{z}_t)$
13:         $\boldsymbol{x}_0 \leftarrow \text{VaeDec}(\boldsymbol{z}_0) \leftarrow \boldsymbol{z}_{t-1}$   // Predict image from original latent
14:         $\mathcal{L} \leftarrow \lambda\phi(\sum_\theta \mathcal{R}_\theta^{k+1}(\boldsymbol{c}_i, \boldsymbol{x}_0))$   // Multi-reward feedback learning loss
15:         $p_{\theta_{i+1}}^k \leftarrow p_{\theta_i}^k$   // Update the base diffusion model
16:     **end for**   // Get $p^{k+1}$ after training
17:     **for** $(\boldsymbol{c}_i, \boldsymbol{x}_0^w, \boldsymbol{x}_0^l) \in \mathcal{D}_k$ **do**
18:         $\boldsymbol{x}_0^* \leftarrow p^{k+1}(\boldsymbol{c}_i)$   // Sample images from optimized base model
19:     **end for**
20:     $\mathcal{D}_{k+1} \leftarrow rank(\mathcal{D}_k \cup \boldsymbol{x}_0^*)$   // Expand the dataset and update ranking
21: **end for**

---

providing more comprehensive guidance to the base diffusion model for compositional generation. The optimization objective of the iterative feedback learning process is formalized in the following lemma (proof provided in the appendix A.2):

**Lemma 1.** *The unified optimization framework of iterative feedback learning can be formulated as:*

$$\max_\theta \; J(\theta) = \mathbb{E}_{\left[\boldsymbol{c} \sim \mathcal{C}, (\boldsymbol{x}_0^w, \boldsymbol{x}_0^l) \sim p_\theta^*(\cdot \mid \boldsymbol{c})\right]} \left[\log \sigma\left(\beta \log \frac{p_\theta^*\left(\boldsymbol{x}_{0:T}^w \mid \boldsymbol{c}\right)}{p_{\text{ref}}\left(\boldsymbol{x}_{0:T}^w \mid \boldsymbol{c}\right)} - \beta \log \frac{p_\theta^*\left(\boldsymbol{x}_{0:T}^l \mid \boldsymbol{c}\right)}{p_{\text{ref}}\left(\boldsymbol{x}_{0:T}^l \mid \boldsymbol{c}\right)}\right)\right] \quad (3)$$

where $p^*(\cdot)$ denotes the optimized base diffusion model. We simplify the bilevel problem of iterative feedback learning into a single-level objective. Based on this, we present the following theorem regarding the gradient of this objective:

**Theorem 1.** *Assume that* $F_\theta(\boldsymbol{c}, \boldsymbol{x}_0^w, \boldsymbol{x}_0^l) = \log \sigma\left(\beta \log \frac{p_\theta^*(\boldsymbol{x}_{0:T}^w \mid \boldsymbol{c})}{p_{\text{ref}}\left(\boldsymbol{x}_{0:T}^w \mid \boldsymbol{c}\right)} - \beta \log \frac{p_\theta^*(\boldsymbol{x}_{0:T}^l \mid \boldsymbol{c})}{p_{\text{ref}}\left(\boldsymbol{x}_{0:T}^l \mid \boldsymbol{c}\right)}\right)$, *the gradient of optimization object can be written as the sum of two terms:* $\nabla_\theta J(\theta) = T_1 + T_2$, *where:*

$$T_1 = \mathbb{E}\left[\left(\nabla_\theta \log p_\theta\left(\boldsymbol{x}_{0:T}^w \mid \boldsymbol{c}\right) + \nabla_\theta \log p_\theta\left(\boldsymbol{x}_{0:T}^l \mid \boldsymbol{c}\right)\right) F_\theta\left(\boldsymbol{c}, \boldsymbol{x}_0^w, \boldsymbol{x}_0^l\right)\right] \quad (4)$$

$$T_2 = \mathbb{E}_{\left[\boldsymbol{c} \sim \mathcal{C}, (\boldsymbol{x}_0^w, \boldsymbol{x}_0^l) \sim p_\theta^*(\cdot \mid \boldsymbol{c})\right]}\left[\nabla_\theta[F_\theta(\boldsymbol{c}, \boldsymbol{x}_0^w, \boldsymbol{x}_0^l)]\right] \quad (5)$$

It is evident that $T_2$ represents the gradient form of direct preference optimization. In addition, we have another term $T_1$, which guides the gradient of optimization objective. As shown in eq. (4), the gradient directs the generation of $\boldsymbol{x}_0^w$ and $\boldsymbol{x}_0^w$ to optimize the implicit reward function $F_\theta(\boldsymbol{c}, \boldsymbol{x}_0^w, \boldsymbol{x}_0^l)$. The gradient term $T_1$ helps the model better distinguish between winning and losing samples, increasing the probability of generating high-quality images while reducing the probability of generating low-quality images. This improves the model's alignment with the reward model's preferences during generation, thereby enhancing the comprehensive capabilities of compositional generation.

**Superiority over Diffusion-DPO and ImageReward**    Here we clarify some superiorities of Iter-Comp over Diffusion-DPO (Wallace et al., 2024) and ImageReward (Xu et al., 2024). Our IterComp first focuses on composition-aware rewards to optimize T2I models for realistic complex generation scenarios, and constructs a powerful model gallery to collect multiple composition-aware model preferences. Then our novel iterative feedback learning framework can effectively achieve progressive self-refinement of both base diffusion model and reward models over multiple iterations.

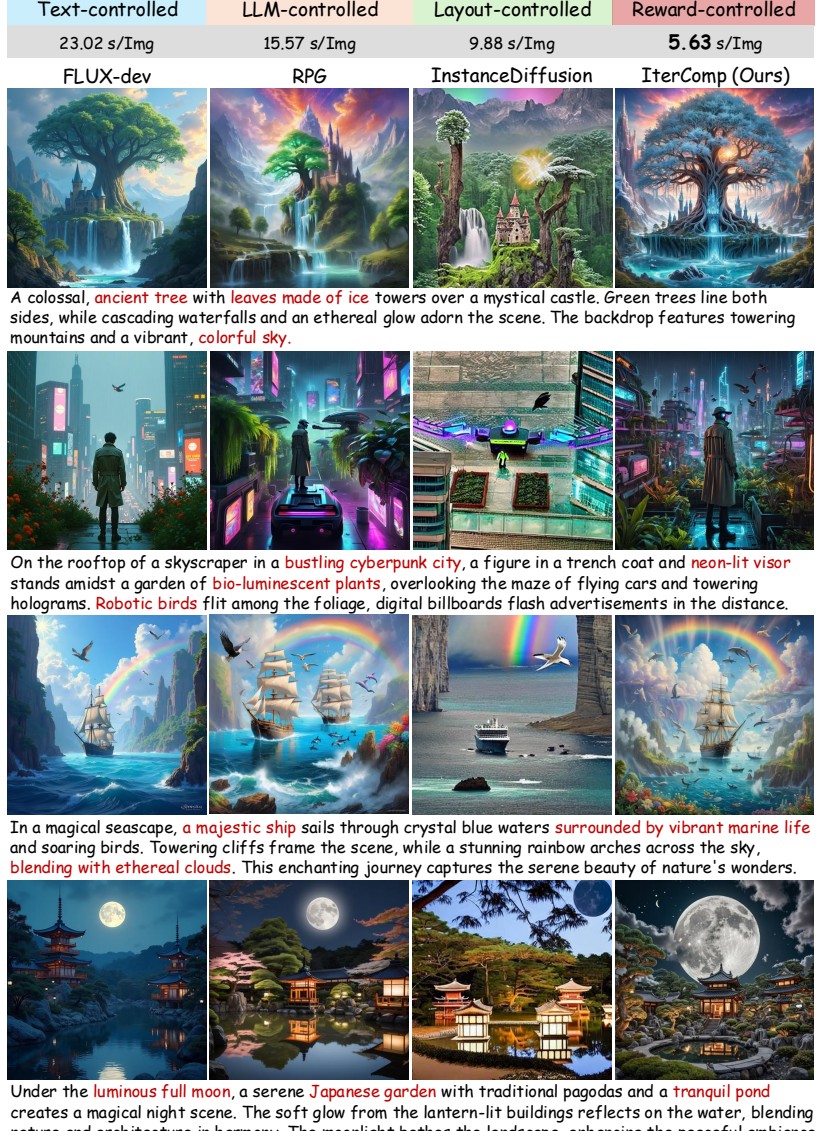

Figure 4: Qualitative comparison between our IterComp and three types of compositional generation methods: text-controlled, LLM-controlled, and layout-controlled approaches. IterComp is the first reward-controlled method for compositional generation, utilizing an iterative feedback learning framework to enhance the compositionality of generated images. Colored text denotes the advantages of IterComp in generated images.

## 4    EXPERIMENTS

**Datasets and Training Setting**    The reward models are trained on the composition-aware model preference dataset, consisting of 3,500 prompts and 52,500 image-rank pairs. For training the three reward models, we finetune BLIP and the learnable MLP with a learning rate of $1e-5$ and a batch size of 64. During the iterative feedback learning process, we randomly select 10,000 prompts from

Table 2: Evaluation results about compositionality on T2I-CompBench (Huang et al., 2023). Iter-Comp consistently demonstrates the best performance regarding attribute binding, object relation-ships, and complex compositions. We denote the best score in blue and the second-best score in green . The baseline data is quoted from GenTron (Chen et al., 2024b).

| Model | Attribute Binding | | | Object Relationship | | Complex↑ |
|---|---|---|---|---|---|---|
| | Color ↑ | Shape↑ | Texture↑ | Spatial↑ | Non-Spatial↑ | |
| Stable Diffusion 1.4 (Rombach et al., 2022) | 0.3765 | 0.3576 | 0.4156 | 0.1246 | 0.3079 | 0.3080 |
| Stable Diffusion 2 (Rombach et al., 2022) | 0.5065 | 0.4221 | 0.4922 | 0.1342 | 0.3096 | 0.3386 |
| Attn-Exct v2 (Chefer et al., 2023) | 0.6400 | 0.4517 | 0.5963 | 0.1455 | 0.3109 | 0.3401 |
| Stable Diffusion XL (Betker et al., 2023) | 0.6369 | 0.5408 | 0.5637 | 0.2032 | 0.3110 | 0.4091 |
| PixArt-$\alpha$ (Chen et al., 2023) | 0.6886 | 0.5582 | 0.7044 | 0.2082 | 0.3179 | 0.4117 |
| ECLIPSE (Patel et al., 2024) | 0.6119 | 0.5429 | 0.6165 | 0.1903 | 0.3139 | - |
| Dimba-G (Fei et al., 2024) | 0.6921 | 0.5707 | 0.6821 | 0.2105 | 0.3298 | 0.4312 |
| GenTron (Chen et al., 2024b) | 0.7674 | 0.5700 | 0.7150 | 0.2098 | 0.3202 | 0.4167 |
| GLIGEN (Li et al., 2023) | 0.4288 | 0.3998 | 0.3904 | 0.2632 | 0.3036 | 0.3420 |
| LMD+ (Lian et al., 2023a) | 0.4814 | 0.4865 | 0.5699 | 0.2537 | 0.2828 | 0.3323 |
| InstanceDiffusion (Wang et al., 2024a) | 0.5433 | 0.4472 | 0.5293 | 0.2791 | 0.2947 | 0.3602 |
| **IterComp (Ours)** | 0.7982 | 0.6217 | 0.7683 | 0.3196 | 0.3371 | 0.4873 |

DiffusionDB (Wang et al., 2022) and use SDXL (Betker et al., 2023) as the base diffusion model, finetuning it with a learning rate of $1e-5$ and a batch size of 4. We set $T=40$, $[T_1, T_2]=[1, 10]$, $\phi = \mathrm{ReLU}$, and $\lambda = 1e-3$. All experiments are conducted on 4 NVIDIA A100 GPUs.

**Baseline Models**   We curate a model gallery of six open-source models, each excelling in different aspects of compositional generation: FLUX (BlackForest, 2024), Stable Diffusion 3 (Esser et al., 2024), SDXL (Betker et al., 2023), Stable Diffusion 1.5 (Rombach et al., 2022), RPG (Yang et al., 2024b), and InstanceDiffusion (Wang et al., 2024a). To ensure the base diffusion model thoroughly and comprehensively learns composition-aware model preferences, we progressively expand the model gallery by incorporating new models (e.g., Omost (Omost-Team, 2024), Stable Cascade (Per-nias et al., 2023), PixArt-$\alpha$ (Chen et al., 2023)) at each iteration. For performance comparison in compositional generation, we select several state-of-the-art methods, including FLUX (BlackForest, 2024), SDXL (Betker et al., 2023), and RPG (Yang et al., 2024b) to compare with our approach. We use GPT-4o (OpenAI, 2024) for the LLM-controlled methods. Additionally, GPT-4o is also employed to infer the layout from the prompt for the layout-controlled methods.

## 4.1   MAIN RESULTS

**Qualitative Comparison**   As shown in fig. 4, IterComp achieves superior compositional genera-tion results compared to the three main types of compositional generation methods: text-controlled, LLM-controlled, and layout-controlled approaches. In comparison to text-controlled methods FLUX (BlackForest, 2024), IterComp excels in handling spatial relationships, significantly reduc-ing errors such as object omissions and inaccuracies in numeracy and positioning. When compared to LLM-controlled methods like RPG (Yang et al., 2024b), IterComp produces more reasonable object placements, avoiding the unrealistic positioning caused by LLM hallucinations. Compared to layout-controlled methods like InstanceDiffusion (Wang et al., 2024a), IterComp demonstrates a clear advantage in both semantic aesthetics and compositionality, particularly when generating under complex prompts.

**Quantitative Comparison**   We compare IterComp with previous outstanding compositional text/layout-to-image models on the T2I-CompBench (Huang et al., 2023) in six key compositional scenarios. As shown in table 2, IterComp demonstrates a remarkable preference across all evalu-ation tasks. Layout-controlled methods such as LMD+ (Lian et al., 2023a) and InstanceDiffusion (Wang et al., 2024a) excel in generating accurate spatial relationships, while text-to-image mod-els like SDXL (Betker et al., 2023) and GenTron (Chen et al., 2024b) exhibit particular strengths in attribute binding and non-spatial relationships. In contrast, IterComp achieves comprehensive improvement in compositional generation. It obtains the strengths of various models by collecting composition-aware model preferences, and employs a novel iterative feedback learning to enable self-refinement of both the base diffusion model and reward models in a closed-loop manner.

IterComp achieves a high level of compositionality while simultaneously enhancing the realism and aesthetics of the generated images. As shown in table 3, we evaluate the improvement in image

Table 3: Evaluation on image realism.

| Model | CLIP Score↑ | Aesthetic Score↑ | ImageReward↑ |
|---|---|---|---|
| Stable Diffusion 1.4 (Rombach et al., 2022) | 0.307 | 5.326 | -0.065 |
| Stable Diffusion 2.1 (Rombach et al., 2022) | 0.321 | 5.458 | 0.216 |
| Stable Diffusion XL (Betker et al., 2023) | 0.322 | 5.531 | 0.780 |
| GLIGEN (Li et al., 2023) | 0.301 | 4.892 | -0.077 |
| LMD+ (Lian et al., 2023a) | 0.298 | 4.964 | -0.072 |
| InstanceDiffusion (Wang et al., 2024a) | 0.302 | 5.042 | -0.035 |
| IterComp (Ours) | 0.337 | 5.936 | 1.437 |

Table 4: Evaluation on inference time.

| Model | Inference Time↓ |
|---|---|
| FLUX-dev | 23.02 s/Img |
| Stable Diffusion XL (Betker et al., 2023) | 5.63 s/Img |
| Omost (Omost-Team, 2024) | 21.08 s/Img |
| RPG (Yang et al., 2024b) | 15.57 s/Img |
| InstanceDiffusion (Wang et al., 2024a) | 9.88 s/Img |
| IterComp (Ours) | 5.63 s/Img |

Table 5: Comparison between IterComp and other diffusion alignment methods.

| Model | Average Result on T2I-CB↑ | CLIP Score↑ | Aesthetic Score↑ |
|---|---|---|---|
| Stable Diffusion XL (Betker et al., 2023) | 0.4441 | 0.322 | 5.531 |
| Diffusion-DPO (Wallace et al., 2024) | 0.4417 | 0.326 | 5.572 |
| ImageReward (Xu et al., 2024) | 0.4639 | 0.323 | 5.613 |
| IterComp (Ours) | 0.5554 | 0.337 | 5.936 |

realism by calculating the CLIP Score, Aesthetic Score, and ImageReward. IterComp significantly outperforms previous models across all three scenarios, demonstrating remarkable fidelity and precision in alignment with the complex text prompt. These promising results highlight the versatility of IterComp in both compositionality and fidelity. We provide more quantitative comparison results between IterComp and other diffusion alignment methods in appendix A.7.

IterComp requires less time to generate high-quality images. In table 4, we compare the inference time of IterComp with other outstanding models, such as FLUX (BlackForest, 2024), RPG (Yang et al., 2024b) in generating a single image. Using the same text prompts and fixing the denoising steps to 40, IterComp demonstrates faster generation, because it avoids the complex attention computations in RPG and Omost. Our method can incorporate composition-aware knowledge from different models without adding any computational overhead. This efficiency highlights its potential for various applications and offers a new perspective on handling complex generation tasks.

We compare IterComp with state-of-the-art diffusion alignment methods, Diffusion-DPO (Wallace et al., 2024) and ImageReward (Xu et al., 2024). As demonstrated in table 5, IterComp significantly outperforms previous diffusion alignment methods across all three scenarios. Iterative feedback learning allows models to achieve self-refinement over multiple iterations, resulting in comprehensive improvements in compositionality and realism.

## 4.2 Ablation Study

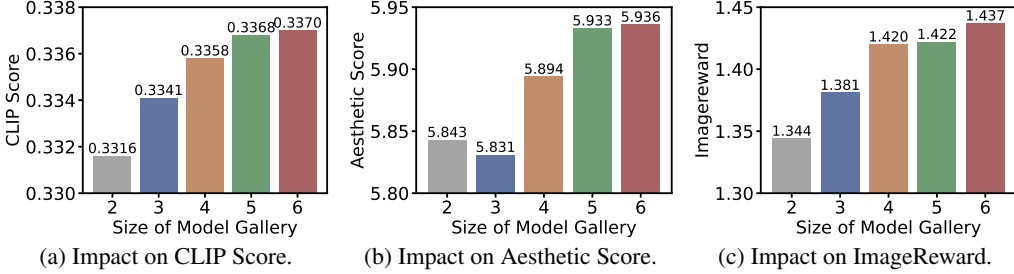

(a) Impact on CLIP Score.    (b) Impact on Aesthetic Score.    (c) Impact on ImageReward.

Figure 5: Ablation study on the model gallery size.

**Effect of Model Gallery Size**  In the ablation study on model gallery size, as shown in fig. 5, we observe that increasing the size of the model gallery leads to improved performance for Iter-Comp across various evaluation tasks. To leverage this finding and provide more fine-grained reward guidance, we progressively expand the model gallery over multiple iterations by incorporating the optimized base diffusion model and new models such as Omost (Omost-Team, 2024).

**Effect of Composition-aware Feedback Learning**  We conducted an ablation study (see fig. 6) to evaluate the impact of composition-aware iterative feedback learning. The results show that this approach significantly improves both the accuracy of compositional generation and the aesthetic quality of the generated images. As the number of iterations increases, the model's preferences gradually converge. Based on this observation, we set the number of iterations to 3 in IterComp.

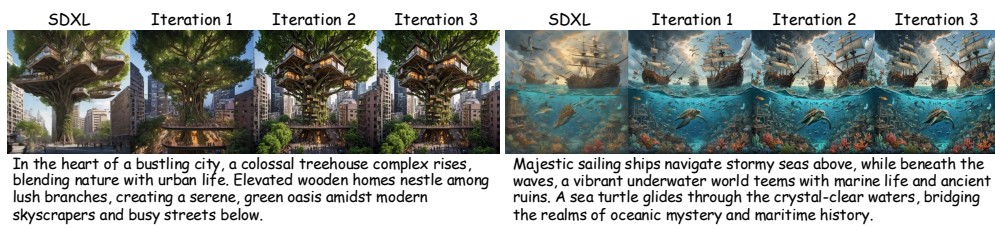

Figure 6: Ablation study on the iterations of feedback learning.

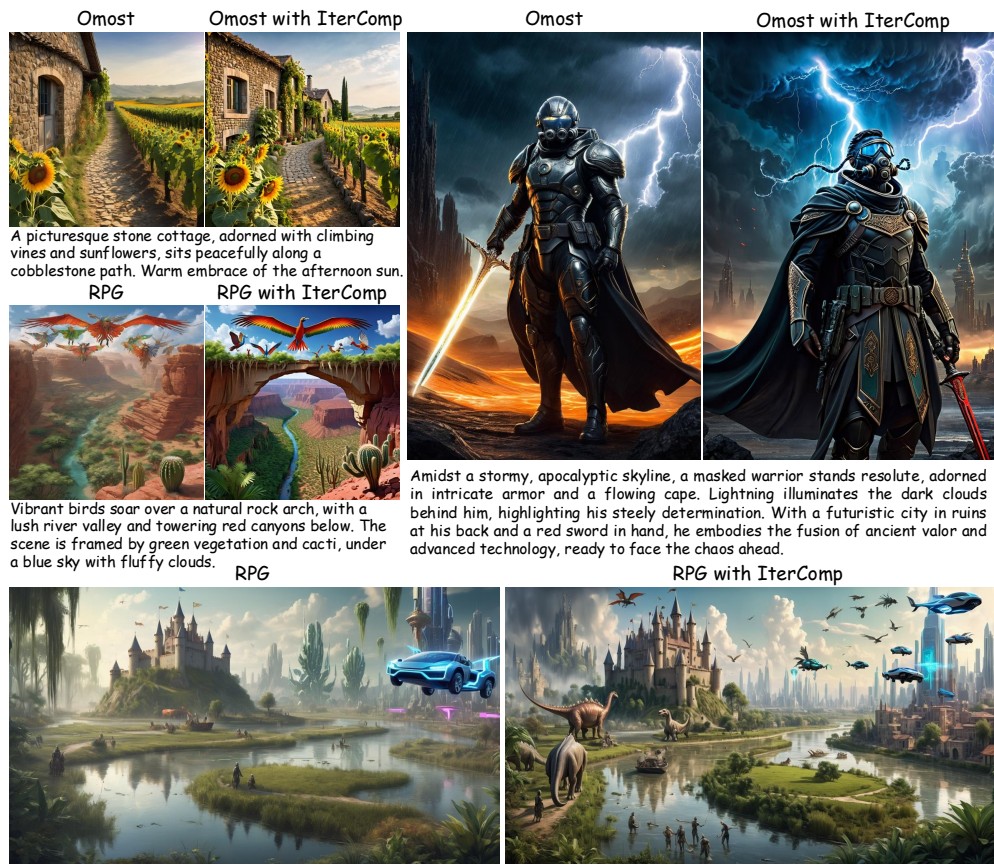

Figure 7: The generation performance of integrating IterComp into RPG and Omost.

**Generalization Study** IterComp can serve as a powerful backbone for various compositional generation tasks, leveraging its strengths in spatial awareness, complex prompt comprehension, and faster inference. As shown in fig. 7, we integrate IterComp into Omost (Omost-Team, 2024) and RPG (Yang et al., 2024b). The results demonstrate that equipped with the more powerful IterComp backbone, both Omost and RPG achieve excellent compositional generation performance, highlighting IterComp's strong generalization ability and potential for broader applications.

## 5 CONCLUSION

In this paper, we propose a novel framework, IterComp, to address the challenges of complex and compositional text-to-image generation. IterComp aggregates composition-aware model preferences from a model gallery and employs an iterative feedback learning approach to progressively refine both the reward models and the base diffusion models over multiple iterations. For future work, we plan to further enhance this framework by incorporating more complex modalities as input conditions and extending it to more practical applications.

ACKNOWLEDGEMENT

This work is supported by National Natural Science Foundation of China (U23B2048, U22B2037), Beijing Municipal Science and Technology Project (Z231100010323002), research grant No. SH2024JK29 and High-performance Computing Platform of Peking University. This work is also supported by the National Key Research and Development Program of China (No. 2024YFB2808903) , the research fund of Tsinghua University - Tencent Joint Laboratory for Internet Innovation Technology.

The author team would like to deliver sincere thanks to Ruihang Chu from Tsinghua University for his significant suggestions for the refinement of this paper.

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

# A APPENDIX

This supplementary material is structured into several sections that provide additional details and analysis related to IterComp. Specifically, it will cover the following topics:

- In appendix A.1, we provide a preliminary about Stable Diffusion (SD) and Reward Feedback Learning (ReFL).
- In appendix A.2, we provide detailed theoretical proof of the effectiveness of iterative feedback learning.
- In appendix A.3, we conduct an experimental analysis to assess model stability.
- In appendix A.4, we provide the results of user study.
- In appendix A.5, we present the quantitative comparison between IterComp and RPG.
- In appendix A.6, we present the quantitative comparison between IterComp and two layout-based models: InstanceDiffusion and MIGC.
- In appendix A.7, we provide more visualization results for IterComp.

## A.1 PRELIMINARY

**Stable Diffusion**    Stable Diffusion (SD) (Rombach et al., 2022) performs multi-step denoising on random noise $z_T \sim \mathcal{N}(\mathbf{0}, \mathbf{I})$ to generate a clear latent $z_0$ in the latent space under the guidance of text prompt $c$. During the training, an input image $x_0$ is processed by a pretrained autoencoder to obtain its latent representation $z_0$. A random noise $\epsilon \sim \mathcal{N}(\mathbf{0}, \mathbf{I})$ is injected into $z_0$ in the forward process as follow:

$$z_t = \sqrt{\bar{\alpha}_t} z_0 + \sqrt{1 - \bar{\alpha}_t} \epsilon \tag{6}$$

where $\alpha_t$ is the noise schedule. The UNet $\epsilon_\theta$ is trained to predict the added noise with the optimization objective:

$$\min_\theta \mathcal{L}(\theta) = \mathbb{E}_{[z_0 \sim \mathcal{E}(x_0), \epsilon \sim \mathcal{N}(\mathbf{0}, \mathbf{I}), t]} \left[ \|\epsilon - \epsilon_\theta(z_t, t, \tau(c))\|_2^2 \right] \tag{7}$$

where $\mathcal{E}(\cdot)$ denote the preteained encoder of VAE, $\tau(\cdot)$ denotes the pretrained text encoder.

**Reward Feedback Learning**    Reward Feedback Learning (ReFL) (Xu et al., 2024) is proposed to align diffusion models with human preferences. The reward model serves as the preference guidance during the finetuning of the diffusion model. ReFL begins with an input prompt $c$ and a random noise $z_T \sim \mathcal{N}(\mathbf{0}, \mathbf{I})$. The noise $z_T$ is progressively denoised until it reaches a randomly selected timestep $t$. The latent $z_0$ is directly predicted from $z_t$, and the decoder from a pretrained VAE is used to generate the predicted image $x_0$. The pretrained reward model $\mathcal{R}(\cdot)$ provides a reward score as feedback, which is used to finetune the diffusion model as follows:

$$\min_\theta \mathcal{L}(\theta) = -\mathbb{E}_{c \sim \mathcal{C}} \left( \mathcal{R}(c, x_0) \right) \tag{8}$$

where the prompt $c$ is randomly selected from the prompt dataset $\mathcal{C}$.

## A.2 THEORETICAL PROOF OF THE EFFECTIVENESS OF ITERATIVE FEEDBACK LEARNING

### A.2.1 PROOF OF LEMMA 1

*Proof of Lemma 1.* Considering the general form of RLHF, we change the optimization problem of iterative feedback learning to a bilevel optimization (Wallace et al., 2024; Ding et al., 2024):

$$\min_{\mathcal{R}} \quad -\mathbb{E}_{[c \sim \mathcal{C}, (x_0^w, x_0^l) \sim p_\mathcal{R}^*(\cdot|c)]} \left[ \log \sigma \left( \mathcal{R}(c, x_0^w) - \mathcal{R}(c, x_0^l) \right) \right]$$
$$\text{s.t. } p_\mathcal{R}^* := \arg \max_p \mathbb{E}_{c \sim \mathcal{C}} \left[ \mathbb{E}_{x_0 \sim p(\cdot|c)} \mathcal{R}(c, x_0) \right] - \beta \mathbb{D}_{\mathrm{KL}}[p(x_{0:T} \mid c) || p_{\mathrm{ref}}(x_{0:T} \mid c)] \tag{9}$$

where $p_\mathcal{R}^*$ denotes the optimized base models under the guidance of reward model $\mathcal{R}$. We have the reparameterization of the reward model (also shown in previous works by (Wallace et al., 2024)):

$$\mathcal{R}(c, x_0) = \beta \mathbb{E}_{p_\mathcal{R}(x_{1:T}|x_0, c)} \left[ \log \frac{p_\mathcal{R}^*(x_{0:T} \mid c)}{p_{\mathrm{ref}}(x_{0:T} \mid c)} \right] + \beta \log Z(c) \tag{10}$$

$$Z(\boldsymbol{c}) = \sum_{\boldsymbol{x}} p_{\text{ref}}\left(\boldsymbol{x}_{0:T} \mid \boldsymbol{c}\right) \exp\left(\mathcal{R}(\boldsymbol{c}, \boldsymbol{x}_0)/\beta\right) \tag{11}$$

Substituting this reward reparameterization into eq. (9), we get the new optimization objective as:

$$\min_{p_{\mathcal{R}}^*} -\mathbb{E}_{\left[\boldsymbol{c}\sim\mathcal{C},(\boldsymbol{x}_0^w,\boldsymbol{x}_0^l)\sim p_{\mathcal{R}}^*(\cdot|\boldsymbol{c})\right]} \left[\log \sigma\left(\beta \log \frac{p_{\mathcal{R}}^*\left(\boldsymbol{x}_{0:T}^w \mid \boldsymbol{c}\right)}{p_{\text{ref}}\left(\boldsymbol{x}_{0:T}^w \mid \boldsymbol{c}\right)} - \beta \log \frac{p_{\mathcal{R}}^*\left(\boldsymbol{x}_{0:T}^l \mid \boldsymbol{c}\right)}{p_{\text{ref}}\left(\boldsymbol{x}_{0:T}^l \mid \boldsymbol{c}\right)}\right)\right] \tag{12}$$

This new optimization objective is denoted as $J(p_{\mathcal{R}}^*)$, we get:

$$\max_{p_{\mathcal{R}}^*} \ J(p_{\mathcal{R}}^*) = \mathbb{E}_{\left[\boldsymbol{c}\sim\mathcal{C},(\boldsymbol{x}_0^w,\boldsymbol{x}_0^l)\sim p_{\mathcal{R}}^*(\cdot|\boldsymbol{c})\right]} \left[\log \sigma\left(\beta \log \frac{p_{\mathcal{R}}^*\left(\boldsymbol{x}_{0:T}^w \mid \boldsymbol{c}\right)}{p_{\text{ref}}\left(\boldsymbol{x}_{0:T}^w \mid \boldsymbol{c}\right)} - \beta \log \frac{p_{\mathcal{R}}^*\left(\boldsymbol{x}_{0:T}^l \mid \boldsymbol{c}\right)}{p_{\text{ref}}\left(\boldsymbol{x}_{0:T}^l \mid \boldsymbol{c}\right)}\right)\right] \tag{13}$$

We use $p_\theta$ to parameterize the policy and formulate the final optimization objective as:

$$\max_{\theta} \ J(\theta) = \mathbb{E}_{\left[\boldsymbol{c}\sim\mathcal{C},(\boldsymbol{x}_0^w,\boldsymbol{x}_0^l)\sim p_\theta^*(\cdot|\boldsymbol{c})\right]} \left[\log \sigma\left(\beta \log \frac{p_\theta^*\left(\boldsymbol{x}_{0:T}^w \mid \boldsymbol{c}\right)}{p_{\text{ref}}\left(\boldsymbol{x}_{0:T}^w \mid \boldsymbol{c}\right)} - \beta \log \frac{p_\theta^*\left(\boldsymbol{x}_{0:T}^l \mid \boldsymbol{c}\right)}{p_{\text{ref}}\left(\boldsymbol{x}_{0:T}^l \mid \boldsymbol{c}\right)}\right)\right] \tag{14}$$

$$\square$$

### A.2.2   PROOF OF THEOREM 1

*Proof of Theorem 1.* The gradient of the optimization objective in eq. (14) can be written as:

$$\nabla_\theta J(\theta) = \nabla_\theta \sum_{\boldsymbol{c},\boldsymbol{x}_0^w,\boldsymbol{x}_0^l} p_\theta(\boldsymbol{x}_{0:T}^w \mid \boldsymbol{c}) \, p_\theta(\boldsymbol{x}_{0:T}^l \mid \boldsymbol{c}) \left[\log \sigma\left(\beta \log \frac{p_\theta^*\left(\boldsymbol{x}_{0:T}^w \mid \boldsymbol{c}\right)}{p_{\text{ref}}\left(\boldsymbol{x}_{0:T}^w \mid \boldsymbol{c}\right)} - \beta \log \frac{p_\theta^*\left(\boldsymbol{x}_{0:T}^l \mid \boldsymbol{c}\right)}{p_{\text{ref}}\left(\boldsymbol{x}_{0:T}^l \mid \boldsymbol{c}\right)}\right)\right] \tag{15}$$

Assume that:

$$F_\theta(\boldsymbol{c}, \boldsymbol{x}_0^w, \boldsymbol{x}_0^l) = \log \sigma\left(\beta \log \frac{p_\theta^*\left(\boldsymbol{x}_{0:T}^w \mid \boldsymbol{c}\right)}{p_{\text{ref}}\left(\boldsymbol{x}_{0:T}^w \mid \boldsymbol{c}\right)} - \beta \log \frac{p_\theta^*\left(\boldsymbol{x}_{0:T}^l \mid \boldsymbol{c}\right)}{p_{\text{ref}}\left(\boldsymbol{x}_{0:T}^l \mid \boldsymbol{c}\right)}\right) \tag{16}$$

$$\hat{p}_\theta\left(\boldsymbol{x}_{0:T}^w, \boldsymbol{x}_{0:T}^l \mid \boldsymbol{c}\right) = p_\theta\left(\boldsymbol{x}_{0:T}^w \mid \boldsymbol{c}\right) p_\theta(\boldsymbol{x}_{0:T}^l \mid \boldsymbol{c}) \tag{17}$$

The gradient can be decomposed into two terms:

$$\begin{aligned}
\nabla_\theta J(\theta) &= \nabla_\theta \sum_{\boldsymbol{c},\boldsymbol{x}_0^w,\boldsymbol{x}_0^l} \hat{p}_\theta\left(\boldsymbol{x}_{0:T}^w, \boldsymbol{x}_{0:T}^l \mid \boldsymbol{c}\right) F_\theta(\boldsymbol{c}, \boldsymbol{x}_0^w, \boldsymbol{x}_0^l) \\
&= \underbrace{\sum_{\boldsymbol{c},\boldsymbol{x}_0^w,\boldsymbol{x}_0^l} \nabla_\theta \hat{p}_\theta\left(\boldsymbol{x}_{0:T}^w, \boldsymbol{x}_{0:T}^l \mid \boldsymbol{c}\right) F_\theta(\boldsymbol{c}, \boldsymbol{x}_0^w, \boldsymbol{x}_0^l)}_{T_1} + \underbrace{\mathbb{E}_{\left[\boldsymbol{c}\sim\mathcal{C},(\boldsymbol{x}_0^w,\boldsymbol{x}_0^l)\sim p_\theta^*(\cdot|\boldsymbol{c})\right]} \left[\nabla_\theta[F_\theta(\boldsymbol{c}, \boldsymbol{x}_0^w, \boldsymbol{x}_0^l)]\right]}_{T_2}
\end{aligned} \tag{18}$$

By expanding the distribution $\hat{p}_\theta$ in $T_1$, a more specific form is obtained:

$$\begin{aligned}
T_1 &= \sum_{\boldsymbol{c},\boldsymbol{x}_0^w,\boldsymbol{x}_0^l} \nabla_\theta \hat{p}_\theta\left(\boldsymbol{x}_{0:T}^w, \boldsymbol{x}_{0:T}^l \mid \boldsymbol{c}\right) F_\theta(\boldsymbol{c}, \boldsymbol{x}_0^w, \boldsymbol{x}_0^l) \\
&= \mathbb{E}\left[\left(\nabla_\theta \log p_\theta\left(\boldsymbol{x}_{0:T}^w \mid \boldsymbol{c}\right) + \nabla_\theta \log p_\theta\left(\boldsymbol{x}_{0:T}^l \mid \boldsymbol{c}\right)\right) F_\theta\left(\boldsymbol{c}, \boldsymbol{x}_0^w, \boldsymbol{x}_0^l\right)\right]
\end{aligned} \tag{19}$$

$$\square$$

### A.3   ANALYSIS ON MODEL STABILITY

To evaluate the model stability, we selected five methods for comparison: SD1.5 (Rombach et al., 2022), SDXL (Podell et al., 2023), InstanceDiffusion (Wang et al., 2024a), Diffusion-DPO (Wallace et al., 2024), and FLUX (BlackForest, 2024), along with two evaluation metrics: Complex and CLIP-score. Using the same 50 seeds, we calculated the mean and variance of the models' performance for these metrics. To facilitate visualization, we used the variance of each method as the radius and scaled it uniformly by a common factor $(10^4)$ for stability analysis.

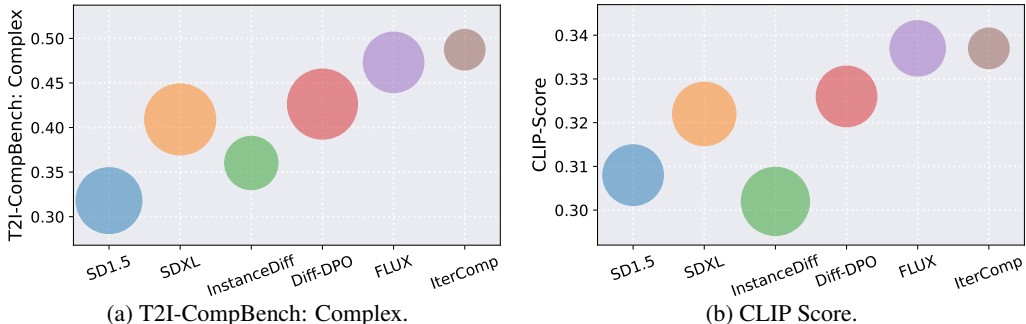

(a) T2I-CompBench: Complex.         (b) CLIP Score.

Figure 8: Analysis on model stability.

Regarding the stability of compositionality, as shown in fig. 8a, we found that IterComp not only achieved the best overall performance but also demonstrated superior stability. This can be attributed to the iterative feedback learning paradigm enable the model to analyze and refine its output at each optimization step, effectively self-correcting and self-improving. The iterative training approach enables the model to perform feedback training based on its own generated samples rather than solely relying on external data, this enables the model to steadily improve over multiple iterations based on its own foundation. This enables the model to steadily improve over multiple iterations, building on its existing foundation, which significantly enhances its stability.

For the stability of realism or generation quality, as shown in fig. 8b, our method also exhibited the highest stability. Therefore, the iterative training approach not only improves the model's performance but also substantially enhances its stability across different dimensions.

## A.4 USER STUDY

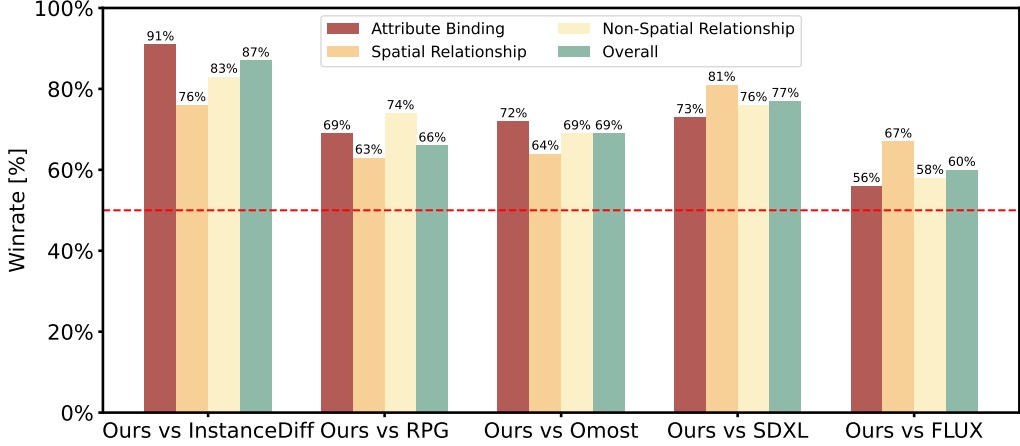

Figure 9: Results of user study.

We conducted a comprehensive user study to evaluate the effectiveness of IterComp in compositional generation. The study involved 41 randomly selected participants from diverse backgrounds. We compared IterComp with five other methods across four aspects: attribute binding, spatial relationships, non-spatial relationships and overall performance. Each comparison involved 25 prompts, culminating in a final survey of 125 prompts and generating 20,500 votes. From the win rate distribution of IterComp shown in the fig. 9, it is evident that IterComp demonstrates significant advantages across all three aspects of compositional generation.

Specifically, compared to the layout-based model InstanceDiffusion (Wang et al., 2024a), IterComp shows an absolute advantage in attribute binding. For text-based models SDXL (Podell et al., 2023) and FLUX (BlackForest, 2024), IterComp leads significantly in spatial relationships. This highlights that the model gallery design effectively collects composition-aware model preferences and enhances performance across different compositional aspects through iterative feedback learning.

Table 6: Comparison between IterComp and RPG on DPG-Bench

| Model | Global | Entity | Attribute | Relation | Other | Average |
|---|---|---|---|---|---|---|
| IterComp | 89.91 | 88.64 | 86.73 | 84.77 | 89.74 | 81.17 |
| RPG | 91.01 | 87.39 | 84.53 | 87.92 | 89.84 | 81.28 |
| RPG+IterComp | 92.74 | 91.33 | 89.10 | 92.38 | 90.13 | 84.72 |

Table 7: Comparison between IterComp and RPG on Genval.

| Model | Single Obj. | Two Obj. | Counting | Colors | Position | Color Attri. | Overall |
|---|---|---|---|---|---|---|---|
| IterComp | 0.97 | 0.85 | 0.63 | 0.86 | 0.33 | 0.41 | 0.675 |
| RPG | 0.97 | 0.86 | 0.66 | 0.79 | 0.30 | 0.38 | 0.660 |
| RPG+IterComp | 0.99 | 0.90 | 0.72 | 0.90 | 0.35 | 0.48 | 0.723 |

## A.5 COMPARISON BETWEEN ITERCOMP AND RPG

We employed two up-to-date benchmarks: DPG-Bench (Hu et al., 2024) and GenEval (Ghosh et al., 2024) for testing to evaluate the capabilities of IterComp and RPG (Yang et al., 2024b) in compositional generation. As demonstrated in table 6 and table 7, IterComp outperforms RPG in metrics like attributes and colors. This is due to our training of a specific reward model for attribute binding, which iteratively enhances IterComp over multiple iterations. Leveraging the strong planning and reasoning capabilities of LLMs, RPG excels in areas such as relations, counting, and positioning. When IterComp is used as the backbone for RPG, the model exhibits remarkable performance across all aspects. This highlights IterComp's superiority in compositional generation. It's important to note that IterComp is a simple SDXL-like model that doesn't require complex computations during inference. As a result, under the same conditions such as prompts and inference steps, IterComp is nearly three times faster than RPG.

## A.6 COMPARISON BETWEEN ITERCOMP AND LAYOUT-BASED METHODS

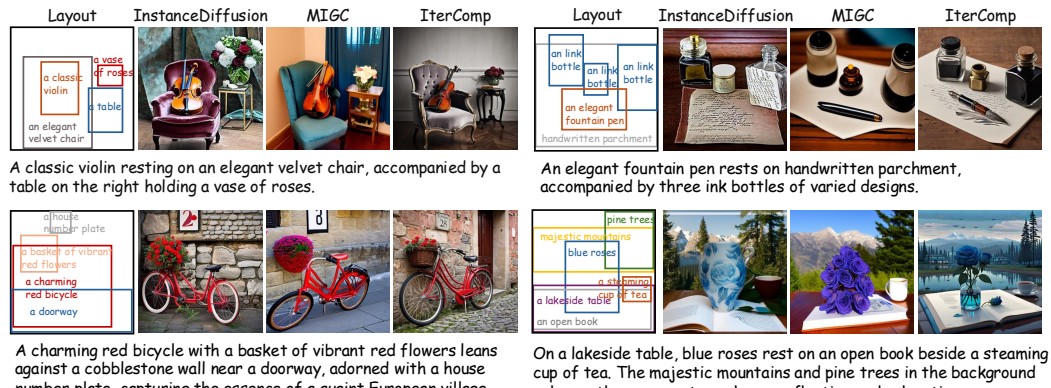

A classic violin resting on an elegant velvet chair, accompanied by a table on the right holding a vase of roses.

An elegant fountain pen rests on handwritten parchment, accompanied by three ink bottles of varied designs.

A charming red bicycle with a basket of vibrant red flowers leans against a cobblestone wall near a doorway, adorned with a house number plate, capturing the essence of a quaint European village.

On a lakeside table, blue roses rest on an open book beside a steaming cup of tea. The majestic mountains and pine trees in the background enhance the serene atmosphere, reflection and relaxation.

Figure 10: Qualitative comparison between IterComp and two layout-to-image methods: InstanceDiffusion and MIGC.

We provide additional experiments between IterComp, InstanceDiffusion (Wang et al., 2024a), and MIGC (Zhou et al., 2024). As shown in fig. 10, these examples clearly show that while MIGC and InstanceDiffusion can accurately generate objects in the specified positions of the layout, there is a notable gap in generation quality compared to IterComp, such as aesthetics and details. Moreover, the images generated by these two methods often appear visually unrealistic, with significant flaws such as incomplete violins or mismatches between bicycle and its basket. This highlights the clear superiority of our IterComp on compositional generaton.

## A.7 More Visualization Results

| Text-controlled | LLM-controlled | Layout-controlled | Reward-controlled |
|---|---|---|---|
| 23.02 s/Img | 15.57 s/Img | 9.88 s/Img | **5.63** s/Img |
| FLUX-dev | RPG | InstanceDiffusion | IterComp (Ours) |

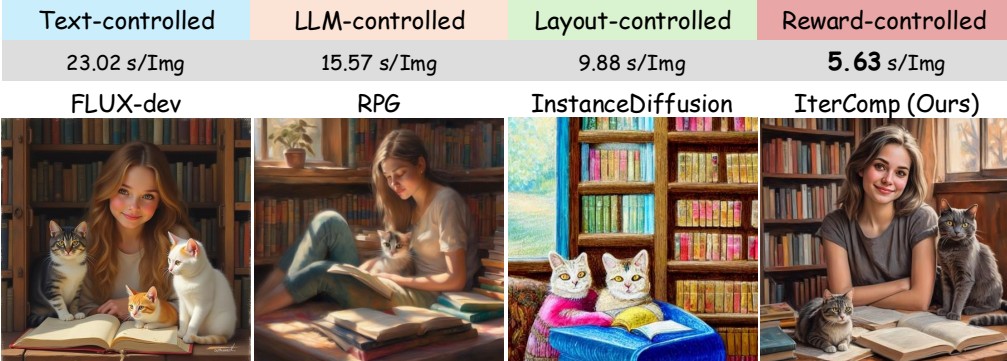

In a cozy library nook, A young woman with expressive, natural eyes and a gentle smile, soft oil brushwork adding warmth and depth to her skin, two curious cats sit beside open books, capturing a moment of quiet companionship. Shelves filled with colorful volumes surround them, and warm sunlight streams through the window, textured chalk pastel for subtle highlights

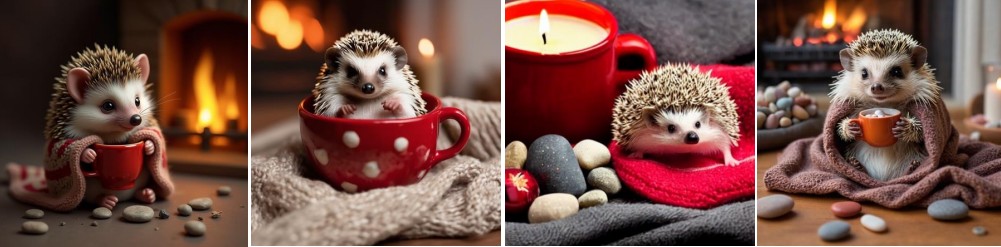

A tiny hedgehog wrapped snugly in a miniature blanket, holding a red cup with both paws, with a fireplace behind, a lit candle on the right, and some scattered pebbles nearby.

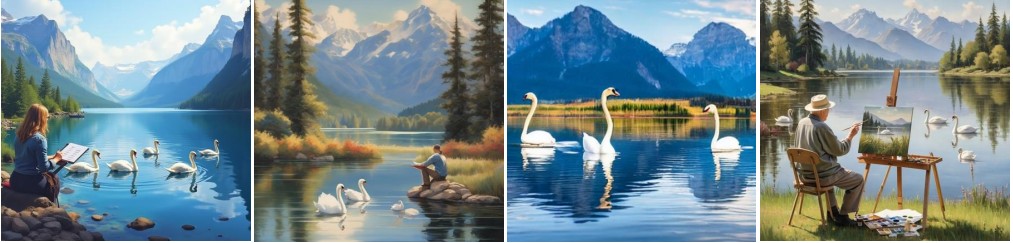

An artist captures the serene beauty of a tranquil lake surrounded by majestic mountains. Four swans glide gracefully across the water, mirroring the peaceful scene on his canvas. The vibrant colors of nature and the artist's focused dedication create a harmonious blend of art and reality in this picturesque setting.

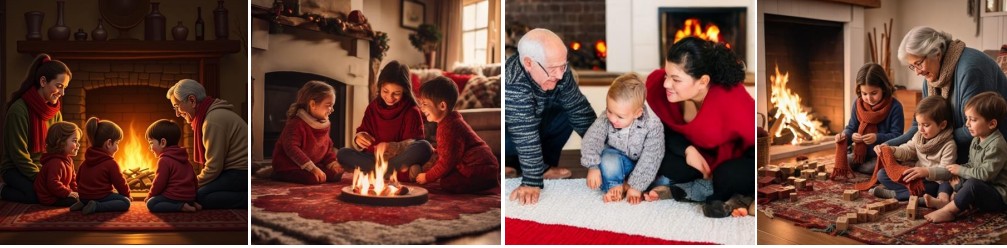

A family gathers by the fireplace, basking in its warm glow. Three children play on the rug while an elderly grandparent watches lovingly. Red scarves and cozy sweaters add to the warmth of the room, complementing the flickering flames of the fire.

Figure 11: Qualitative comparison between IterComp and three types of compositional generation methods: text-controlled, LLM-controlled, and layout-controlled approaches. We use GPT-4o to infer the layout from the prompt for InstanceDiffusion. Colored text denotes the advantages of IterComp in generated images.

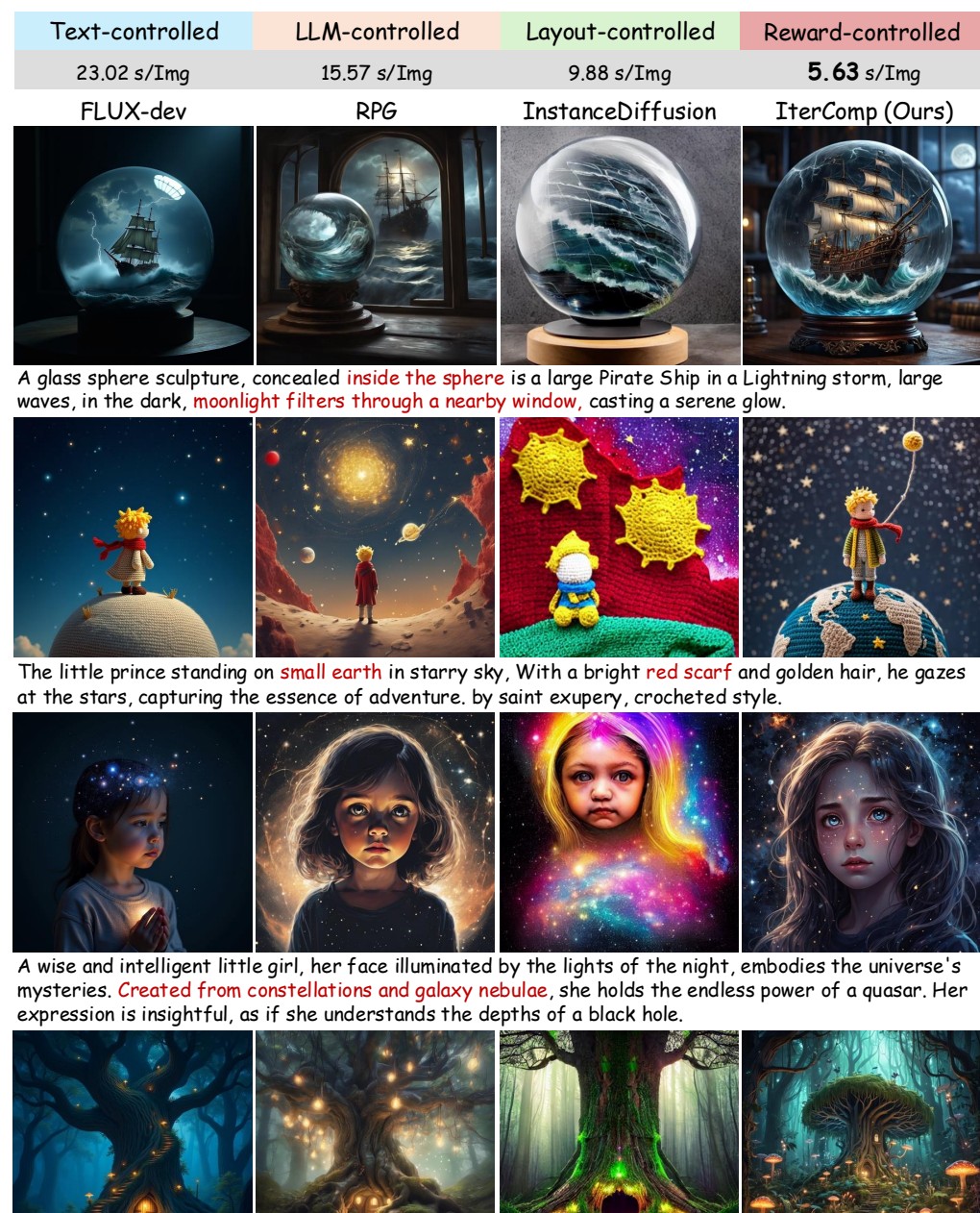

| Text-controlled | LLM-controlled | Layout-controlled | Reward-controlled |
|---|---|---|---|
| 23.02 s/Img | 15.57 s/Img | 9.88 s/Img | **5.63** s/Img |
| FLUX-dev | RPG | InstanceDiffusion | IterComp (Ours) |

A glass sphere sculpture, concealed inside the sphere is a large Pirate Ship in a Lightning storm, large waves, in the dark, moonlight filters through a nearby window, casting a serene glow.

The little prince standing on small earth in starry sky, With a bright red scarf and golden hair, he gazes at the stars, capturing the essence of adventure. by saint exupery, crocheted style.

A wise and intelligent little girl, her face illuminated by the lights of the night, embodies the universe's mysteries. Created from constellations and galaxy nebulae, she holds the endless power of a quasar. Her expression is insightful, as if she understands the depths of a black hole.

In the heart of an enchanted forest, a majestic tree stands illuminated by glowing mushrooms and tiny lights. Its thick roots form a staircase leading to a cozy door, suggesting a hidden world within. The scene is vibrant and magical, inviting wonder and exploration in this mystical woodland realm.

Figure 12: Qualitative comparison between IterComp and three types of compositional generation methods: text-controlled, LLM-controlled, and layout-controlled approaches. We use GPT-4o to infer the layout from the prompt for InstanceDiffusion. Colored text denotes the advantages of IterComp in generated images.

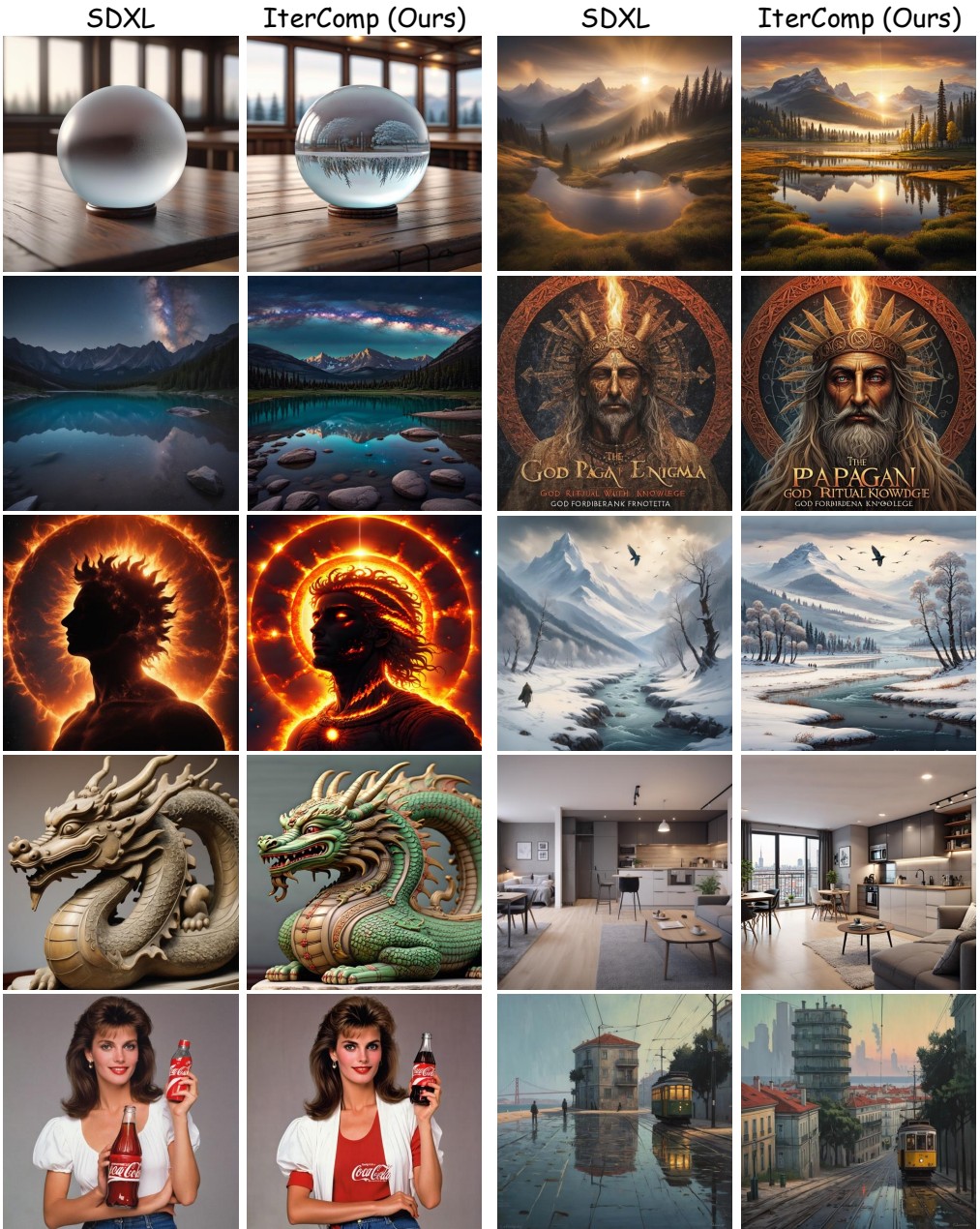

Figure 13: More visualization results for IterComp and its base diffusion model, SDXL.

