# OpenReview forum: "IterComp: Iterative Composition-Aware Feedback Learning from Model Gallery for Text-to-Image Generation"
_ICLR.cc/2025/Conference — ICLR 2025 Poster_

### Official Review · Reviewer_ocHL · 2024-10-23

**Soundness:** 3
**Presentation:** 3
**Contribution:** 3
**Rating:** 8
**Confidence:** 4

**Summary:**

The paper presents a new dataset for compositional generation where experts evaluate the outputs of multiple pre-trained models. The dataset consists of three aspects of compositionality: attribute binding, spatial relationships, and non-spatial relationships, along with 52.5K image-rank pairs, which can be used for training feedback, ranking, or reward models.

The second contribution of the paper is using the collected dataset to train reward models for each of the three key aspects. A multimodal model (BLIP) is used as a feature extractor for both the prompts and the generated images, and the extracted features are projected by MLPs to output the reward. The goal is to predict how good the given image-prompt pairs are by training the model similar to contrastive learning (moving toward winning examples and away from losing examples).

The third contribution is improving the compositional ability of a base model (SDXL is selected but it should not be limited to that) by optimizing it using the trained reward models. The base model is trained to maximize the reward model's output so that its outputs are better aligned with the reward models, which are trying to enforce compositionality. Furthermore, the paper proposes an iterative update mechanism for both reward models and the base model. Reward models are updated to predict the rankings generated by experts while the base model is updated to maximize the outputs of reward models. Through this process, both the base model and reward models are improved for their specific tasks.

**Important note:** This review's technical content and analysis are my original work. Large Language Models were used solely to improve grammar and writing style, without altering any meanings or substantive feedback.

**Strengths:**

* The collected dataset can lead to better models and new research directions. Furthermore, researchers can follow a similar approach to collect their own data. The data can also be used for other RLHF methods, such as diffusion-DPO.
* The iterative feedback mechanism seems to be a novel way to optimize both reward models and the base model.
* The code is shared, which allows reviewers to follow the details of the proposed method.
* The performance gain from IterComp appears significant, as evaluated through user studies, quantitative analysis, and qualitative assessment.

**Weaknesses:**

* Table 5 should be in the main part instead of the Appendix, as it simply demonstrates that the proposed method outperforms previous methods.
* Several experiments are missing:
   - The paper combines all reward models simultaneously, likely leading to improved compositional performance. However, reviewers would benefit from seeing the individual effect of each reward model.
   - While SDXL is chosen as the base model, testing other models would help reviewers understand how reward models affect different base architectures.
   - It would be valuable to examine Diffusion-DPO's performance when trained on the collected dataset. Currently, Diffusion-DPO is trained on the pick-a-pic dataset, which is larger but lacks compositional details. These results would be necessary to evaluate the proposed method's effectiveness using consistent standards.
* Expert ranking may inadvertently include aesthetic information [This observation is meant to prompt discussion rather than highlight a weakness, as the underlying cause remains unclear]:
   - When IterComp is applied, the aesthetic score improves, suggesting that reward models can interpret image aesthetics, since reward maximization leads to better aesthetic scores. This could be because either expert rankings are influenced by image aesthetics (beyond compositional attributes) or because models with better composition naturally generate more aesthetic images (for instance, FLUX, being the best compositional model, likely produces more aesthetically pleasing outputs).
* Some experimental conditions may be misleading:
   - The paper uses 40 inference steps for all models to ensure fairness. However, some models can generate samples with fewer steps; for example, FLUX-dev uses 28 steps by default.

**Questions:**

* Training time considerations:
   - Diffusion-DPO operates in the latent space without requiring image decoding. In contrast, IterComp requires image decoding for the reward models (though this could be avoided by training the reward model with latent space inputs), likely resulting in slower training. Additional commentary on the training scheme would be valuable.
* Potential code and paper discrepancy:
   - The paper describes an iterative feedback mechanism that optimizes both reward models and the base model. However, examination of `feedback_train.py` reveals that only unet parameters (base model) are passed to the optimizer. This suggests that only the base model is being optimized, while reward models remain static. This difference requires clarification.
* Question regarding test-time adaptation:
   - Could the iterative feedback mechanism be applied as test-time adaptation of the base model? Similar to Slot-TTA [1], the base model could be optimized using reward models to improve compositional quality. The process would work as follows: for a given prompt, the base model generates an image, which is then evaluated by reward models. The base model's parameters would be updated to maximize these rewards. This process could be repeated for several iterations. This approach would eliminate the need for training the base model, allowing it to adapt to any prompt at test time through multiple iterations. Comments on this possibility would be valuable.

[1] Test-time Adaptation with Slot-Centric Models, Prabhudesai et al., ICML 2023

EDIT: After the rebuttal and discussion with the authors, I raise my score to 8 from 6.

---

> ### Author Response · Authors · 2024-11-21
> **Response to Reviewer ocHL (Part 1/4)**
>
> *We sincerely thank you for your time and efforts in reviewing our paper, and your valuable feekback. We are glad to see that our method is novel and will lead to new research directions, we provide detailed code to foster community progress, and the experimental results are extensive and promising. Please see below for our responses to your comments.*
>
> **Q1: Table 5 should be in the main part instead of the Appendix, as it simply demonstrates that the proposed method outperforms previous methods.**
>
> A1: Thank you for your suggestion! **In the updated paper**, we have included Table 5 in the main part of the paper.
>
> **Q2: The paper combines all reward models simultaneously, likely leading to improved compositional performance. However, reviewers would benefit from seeing the individual effect of each reward model.**
>
> A2: It's a good suggestion to test the individual effect of each reward model. We used the same training strategy as IterComp but trained with only one reward model at a time. The final results are shown below:
>
> | Model| Color$\uparrow$ | Shape$\uparrow$ | Texture$\uparrow$ | Spatial$\uparrow$ | Non-Spatial$\uparrow$ | Complex$\uparrow$ |
> | --------------------- | --------------- | --------------- | ----------------- | ----------------- | --------------------- | ----------------- |
> | SDXL| 0.6369| 0.5408| 0.5637| 0.2032| 0.3110 |0.4091|
> | IterComp(Attribute)| 0.7742 | 0.6079 | 0.7572| 0.2181 | 0.3129| 0.4429|
> | IterComp(Spatial)| 0.6521| 0.5477 | 0.5881| 0.3008 | 0.3201 |0.4534|
> | IterComp(Non-spatial) | 0.6463| 0.5426  | 0.5724| 0.2265 | 0.3354| 0.4367  |
> | IterComp(All)| **0.7982**      | **0.6217**      | **0.7683**   | **0.3196** | **0.3371**| **0.4873**        |
>
> As shown in the table, training with a single reward model allows the final diffusion model to achieve significant improvements in the specific preference targeted by that reward. Since different preferences are not entirely mutually exclusive, a single reward can also lead to minor enhancements in other metrics. For example, the spatial-based reward model primarily strengthens spatial-awareness  but also demonstrates some degree of non-spatial understanding. Take the scenario of “a person looking at the moon in the sky” as an example: the spatial-based reward model focuses on learning the spatial concept that “the moon is above the person.” At the same time, due to variations in the capabilities of the models in the model gallery, it may also learn the non-spatial relationship of “looking.” Therefore, using only the spatial-based reward model for feedback learning can also improve the model's ability to understand non-spatial relationships.
>
> This indicates that rewards are not entirely independent. For complex tasks such as compositional generation, leveraging multiple reward models in combination can result in more substantial overall improvements.
>
> **Q3: While SDXL is chosen as the base model, testing other models would help reviewers understand how reward models affect different base architectures.**
>
> A3: Thank you for your valuable suggestions and feedback. We applied the same method to perform iterative optimization on both SD1.5 and SD2.1, and the final results are as follows:
>
> | Model            | Color$\uparrow$ | Shape$\uparrow$ | Texture$\uparrow$ | Spatial$\uparrow$ | Non-Spatial$\uparrow$ | Complex$\uparrow$ |
> | ---------------- | --------------- | --------------- | ----------------- | ----------------- | --------------------- | ----------------- |
> | SD1.5            | 0.3821          | 0.3543          | 0.4196            | 0.1277            | 0.3072                | 0.3091            |
> | Itercomp(SD 1.5) | **0.5223**      | **0.3929**      | **0.4890**        | **0.1849**        | **0.3117**            | **0.3590**        |
> | SD2.1            | 0.5044          | 0.4315          | 0.4927            | 0.1365            | 0.3123                | 0.3438            |
> | Itercomp(SD 2.1) | **0.6294**      | **0.4933**      | **0.6072**        | **0.2399**        | **0.3214**            | **0.4177**        |
> | SDXL             | 0.6369          | 0.5408          | 0.5637            | 0.2032            | 0.3110                | 0.4091            |
> | IterComp(SDXL)   | **0.7982**      | **0.6217**      | **0.7683**        | **0.3196**        | **0.3371**            | **0.4873**        |
>
>
> We found that applying IterComp to optimize all three base diffusion models significantly improved compositional generation. Notably, for spatial relationships, our method uses composition-aware reward models to enhance spatial awareness and iteratively improve understanding of positional relationships and quantities. In contrast, base models struggle to develop strong spatial capabilities relying solely on text guidance. Due to time and computational resource constraints, we focused on SD1.5 and SD2.1. Moving forward, we plan to continue refining IterComp and apply it to more advanced models such as FLUX.

---

> ### Author Response · Authors · 2024-11-21
> **Response to Reviewer ocHL (Part 2/4)**
>
> **Q4: It would be valuable to examine Diffusion-DPO's performance when trained on the collected dataset. Currently, Diffusion-DPO is trained on the pick-a-pic dataset, which is larger but lacks compositional details. These results would be necessary to evaluate the proposed method's effectiveness using consistent standards.**
>
> A4: Since Diffusion-DPO lacks an explicitly trained reward model, it cannot accurately rank the images generated by its optimized base model relative to those in the original model gallery. As a result, Diffusion-DPO is unable to perform iterative feedback learning. Therefore, we only trained Diffusion-DPO for a single round using the original dataset and compared its performance against IterComp after first round of training:
>
> | Model                      | Color$\uparrow$ | Shape$\uparrow$ | Texture$\uparrow$ | Spatial$\uparrow$ | Non-Spatial$\uparrow$ | Complex$\uparrow$ |
> | -------------------------- | --------------- | --------------- | ----------------- | ----------------- | --------------------- | ----------------- |
> | SDXL                       | 0.6369          | 0.5408          | 0.5637            | 0.2032            | 0.3110                | 0.4091            |
> | Diffusion-DPO(first round) | 0.6822          | 0.5691          | 0.6224            | 0.2216            | 0.3291                | 0.4263            |
> | IterComp(first round)      | 0.7239          | 0.6083          | 0.6940            | 0.2692            | 0.3308                | 0.4572            |
> | IterComp                   | **0.7982**      | **0.6217**      | **0.7683**        | **0.3196**        | **0.3371**            | **0.4873**        |
>
> From the results, it is evident that IterComp effectively leverages the model gallery to collect composition-aware model preferences, enabling it to learn diverse compositional dimensions during the first round of training. Notably, attributes such as color, texture, and spatial relationships exhibit significant improvement. Compared to Diffusion-DPO, the lack of an explicitly trained reward model leads to mixed training on image pairs with differing preferences, which weakens the model's ability to learn specific compositional preferences. For instance, its improvements in spatial relationships are notably limited, highlighting that the DPO approach is not well-suited for complex tasks like compositional generation.  Moreover, IterComp’s iterative optimization approach delivers substantial advantages across all metrics, further underscoring the effectiveness of iterative feedback learning in addressing complex compositional tasks.
>
> **Q5: Expert ranking may inadvertently include aesthetic information [This observation is meant to prompt discussion rather than highlight a weakness, as the underlying cause remains unclear]: When IterComp is applied, the aesthetic score improves, suggesting that reward models can interpret image aesthetics, since reward maximization leads to better aesthetic scores. This could be because either expert rankings are influenced by image aesthetics (beyond compositional attributes) or because models with better composition naturally generate more aesthetic images (for instance, FLUX, being the best compositional model, likely produces more aesthetically pleasing outputs).**
>
> A5: This is a valuable question. We would like to emphasize that the aesthetic predictor was trained using the Aesthetic Visual Analysis (AVA) and Simulacra Aesthetic Captions (SAC) datasets, which are annotated by experts or photography enthusiasts. It is important to note that these annotators tend to be more critical of images with unrealistic or awkward compositions. The aesthetic predictor is designed to filter for high-quality images that excel in both detail representation and composition. Thus, **reasonable composition is a key aspect of aesthetics, and models with better composition naturally generate more aesthetically pleasing images**. Through multiple iterations of self-refinement, IterComp achieves highly reasonable compositions, resulting in a high degree of aesthetic quality.
>
>
> **Q6: The paper uses 40 inference steps for all models to ensure fairness. However, some models can generate samples with fewer steps; for example, FLUX-dev uses 28 steps by default.**
>
> A6: Through extensive testing, we found that models like IterComp, FLUX, and SDXL are highly robust to the number of inference steps, with their performance being minimally affected. However, InstanceDiffusion is more sensitive to the number of steps. Therefore, for metric evaluation and visualization, we used 20 steps for the other models but 40 steps for InstanceDiffusion. To ensure a fair comparison in terms of inference speed, we set the number of inference steps to 40 for all models. However, this does not affect IterComp's ability to generate higher-quality images at a faster speed compared to other methods.

---

> ### Author Response · Authors · 2024-11-21
> **Response to Reviewer ocHL (Part 3/4)**
>
> **Q7: Diffusion-DPO operates in the latent space without requiring image decoding. In contrast, IterComp requires image decoding for the reward models (though this could be avoided by training the reward model with latent space inputs), likely resulting in slower training. Additional commentary on the training scheme would be valuable.**
>
> A7: Here is an approximate listing of the training time for each stage (all our experiments were conducted on 4*NVIDIA A100-80G GPUs):
>
> | Phrase                                   | Training Time |
> | ---------------------------------------- | ------------- |
> | Train reward models (iteration1)         | 31min         |
> | Train base diffusion models (iteration1) | 3h 37min      |
> | Train reward models (iteration2)         | 14min         |
> | Train base diffusion models (iteration2) | 1h 31min      |
> | Train reward models (iteration3)         | 17min         |
> | Train base diffusion models (iteration3) | 1h 31min      |
> | Total                                    | ~7h 41min     |
>
> To prevent overfitting and to improve the self-correction of the reward model, from the second iteration, we only fine-tuned the reward model using image pairs that include newly generated samples. For example, if the newly generated image is $x_0$, the reward model is fine-tuned on datasets such as $(x_0, x_{SDXL}), (x_0, x_{FLUX})$, while excluding repeated training on pairs like $(x_{SDXL}, x_{FLUX})$. This approach effectively prevents overfitting during multiple iterations while enabling the reward model to better learn composition-aware model preferences. Moreover, from the second iteration, the optimization of base diffusion models reduces the number of epochs by half, from 2 to 1. As a result, with the continuous refinement of iterative feedback learning, the training time of the model  decreases progressively. Ultimately, the total training time is approximately 7 hours and 41 minutes.
>
> **Q8: The paper describes an iterative feedback mechanism that optimizes both reward models and the base model. However, examination of `feedback_train.py` reveals that only unet parameters (base model) are passed to the optimizer. This suggests that only the base model is being optimized, while reward models remain static. This difference requires clarification.**
>
> A8: Thank you for your thorough review. We apologize for any confusion caused. In our codebase, the training of reward models and the base diffusion model **are conducted separately**. The training code for the reward models is located in `train/train.py`, while the training code for the base diffusion model can be found in `Iterative_feedback/feedback_train.py`.
>
> For the iterative feedback learning process, we first expand the model gallery based on the previously optimized diffusion model using `data/iterative_expand_gallery.py`. Then, we enhance the reward models using the expanded model gallery in `train/train.py`. Finally, we fine-tune the base diffusion model using the improved reward model in `Iterative_feedback/feedback_train.py`.
>
> We regret any misunderstanding this has caused and will update our code to consolidate the training of the reward model and the base diffusion model into a single file. This change will streamline our process, eliminate arbitrary filenames, and reduce code complexity. Thank you for your feedback!

---

> ### Author Response · Authors · 2024-11-21
> **Response to Reviewer ocHL (Part 4/4)**
>
> **Q9: Could the iterative feedback mechanism be applied as test-time adaptation of the base model? Similar to Slot-TTA [1], the base model could be optimized using reward models to improve compositional quality. The process would work as follows: for a given prompt, the base model generates an image, which is then evaluated by reward models. The base model's parameters would be updated to maximize these rewards. This process could be repeated for several iterations. This approach would eliminate the need for training the base model, allowing it to adapt to any prompt at test time through multiple iterations. Comments on this possibility would be valuable.**
>
> A9: Thank you for your insightful question. Test-time adaptation is indeed a promising area for advancing diffusion models. Regarding the application of our IterComp framework to this process:
>
> * Iterative Enhancement: In IterComp, images generated by the optimized base model are incorporated back into the reward dataset for ongoing refinement of the reward models. While this iterative process does introduce additional inference time, its potential to significantly enhance generative quality without the need for retraining the base model makes it a worthwhile endeavor in scenarios where inference time is less critical.
>
> * Limitations in Complex Scenarios: It's important to note that relying solely on reward-guided optimization may not suffice in more complex generative tasks, such as chip design, where robust initial models are crucial due to a lack of sufficient training data. In our evaluations within the text-to-image domain, powerful underlying models (e.g., SDXL) ensure a baseline level of quality. If the foundational model lacks basic generation strength, test-time adaptation alone may not yield substantial improvements.
>
> These considerations highlight the potential and limitations of applying IterComp as a test-time adaptation tool, depending on the specific requirements and constraints of the application scenario.

---

> > ### Comment · Reviewer_ocHL · 2024-11-22
> > **Response to authors' response**
> >
> > Thank you for your comprehensive response and the additional experimental results.
> >
> > The new experiments effectively demonstrate IterComp's superior performance compared to baselines like Diffusion-DPO, while also highlighting how each reward type makes its own valuable contribution. Based on your thorough answers and clarifications, I am convinced of the paper's value and am raising my score to 8.
> >
> > I appreciate this fruitful discussion.

---

> ### Author Response · Authors · 2024-11-22
> **Thank you for your support**
>
> Dear Reviewer ocHL，
>
> Many thanks for raising score! We sincerely appreciate your valuable comments and your precious time in reviewing our paper!
>
> Warm Regards,
>
> The Authors

---

### Official Review · Reviewer_bbqi · 2024-10-27

**Soundness:** 3
**Presentation:** 3
**Contribution:** 2
**Rating:** 8
**Confidence:** 4

**Summary:**

This study addresses improving the compositional text-to-image generation capability of less powerful diffusion models. The authors contribute in two main areas. First, they decompose the capabilities of various diffusion models into three domains—attribute binding, spatial relationships, and non-spatial relationships—and rank model outputs accordingly to guide reinforcement learning with reward models specific to each domain. Second, they introduce an iterative training process that utilizes the fine-tuned diffusion model outputs to progressively enhance the reward models.

Through multiple experiments, the study demonstrates the proposed method’s effectiveness, enabling early-stage models to achieve comparable generative abilities with reduced inference time. The authors also verify the effectiveness and general applicability of each design component across different models.

**Strengths:**

1. The paper is well-structured, making it accessible and easy for readers to follow.
2. Clear formulas are provided for each component, effectively removing any ambiguity.
3. Mathematical proofs substantiate the validity of the proposed method.
4. The authors conduct detailed, extensive experiments to support their approach.
5. Illustrative images are included, enhancing clarity and understanding.

**Weaknesses:**

1. About the number of texts used for attribute binding, "500 prompts" in Line 185 is inconsistent with "1500" in Table 1. Which is correct?
2. Although the experiments are detailed, some comparisons appear incomplete. The reinforcement learning from human feedback (RLHF) approach leverages outputs from advanced models like FLUX and SD3 for training, yet direct comparisons with these models are not provided. Including these comparisons would better highlight the method's effectiveness.
3. An additional experiment focusing on the first-round fine-tuned model—trained solely with human-annotated data—would be valuable. This would clarify the necessity and impact of the iterative training approach.

**Questions:**

In addition to the weakness mentioned above, I have a question regarding stability.

Based on my experience, RLHF in diffusion models can often be unstable. I’m curious whether your method consistently produces stable results or if there’s a risk of occasionally poorer outcomes. I’m concerned that the iterative training process might lead to overfitting on biases present in the reward models, potentially reducing overall robustness.

I hope the authors can make up for the weaknesses mentioned and address these questions.

---

> ### Author Response · Authors · 2024-11-21
> **Response to Reviewer bbqi (Part 1/2)**
>
> *We sincerely thank you for your time and efforts in reviewing our paper, and your valuable feekback. We are glad to see that our paper is well-structured and easy to follow, the theoretical details and proof are complete, and the experimental results are extensive and promising. Please see below for our responses to your comments.*
>
> **Q1: About the number of texts used for attribute binding, "500 prompts" in Line 185 is inconsistent with "1500" in Table 1. Which is correct?**
>
> A1: We apologize for the confusion. Attribute binding consists of three aspects: color, shape, and texture. For each aspect, we collect 500 representative prompts, totaling 500*3=1500 prompts for attribute binding. We have revised the problematic expression and **added explanations in the updated pape in line187**. We apologize again for any misunderstandings caused.
>
>
> **Q2: Although the experiments are detailed, some comparisons appear incomplete. The reinforcement learning from human feedback (RLHF) approach leverages outputs from advanced models like FLUX and SD3 for training, yet direct comparisons with these models are not provided. Including these comparisons would better highlight the method's effectiveness.**
>
> A2: Thank you for your suggestion. We've added a comparison of IterComp with FLUX and SD3 regarding compositional aspects as follows:
>
> | Model      | Color$\uparrow$ | Shape$\uparrow$ | Texture$\uparrow$ | Spatial$\uparrow$ | Non-Spatial$\uparrow$ | Complex$\uparrow$ |
> | ---------- | --------------- | --------------- | ----------------- | ----------------- | --------------------- | ----------------- |
> | SDXL       | 0.6369          | 0.5408          | 0.5637            | 0.2032            | 0.3110                | 0.4091            |
> | SD3-medium | 0.7442          | 0.6053          | 0.7055            | 0.2419            | 0.3294                | 0.4548            |
> | FLUX-dev   | 0.7881          | **0.6529**      | 0.7472            | 0.2606            | **0.3396**            | 0.4731            |
> | IterComp   | **0.7982**      | 0.6217          | **0.7683**        | **0.3196**        | 0.3371                | **0.4873**        |
>
> The comparison of IterComp with FLUX and SD3 regarding generation quality and speed as follows:
>
> | Model      | CLlP Score$\uparrow$ | Aesthetic Score$\uparrow$ | ImageReward$\uparrow$ | Inference Time$\downarrow$ |
> | ---------- | -------------------- | ------------------------- | --------------------- | -------------------------- |
> | SDXL       | 0.322                | 5.531                     | 0.780                 | **5.63 s/Img**             |
> | SD3-medium | 0.332                | 5.874                     | 0.942                 | 13.44 s/Img                |
> | FLUX-dev   | **0.339**            | 5.901                     | 1.082                 | 23.02 s/Img                |
> | IterComp   | 0.337                | **5.936**                 | **1.437**             | **5.63 s/Img**             |
>
> From the two tables above, it is clear that IterComp outperforms SD3 in both compositionality and generation quality, with nearly **three times increase in inference speed**. IterComp is on par with FLUX-dev but has a significant advantage in spatial relationships due to our method's iterative enhancement of spatial awareness through a dedicated reward model. **Despite FLUX-dev having 12 billion parameters, our model achieves comparable or superior generation performance with only 1/5 of the parameters and nearly four times faster inference speed.**
>
> It is worth noting that the original IterComp in our paper is optimized based on SDXL, and our main focus is to illustrate how this new optimization framework improves existing diffusion models. Thus it is unfair to make a direct comparison with models like SD3 and FLUX. IterComp is a versatile text-to-image or diffusion alignment framework that can be adapted to any model include FLUX. In the future, we plan to train IterComp based on FLUX, which is expected to demonstrate even more powerful generative capabilities.

---

> ### Author Response · Authors · 2024-11-21
> **Response to Reviewer bbqi (Part 2/2)**
>
> **Q3: An additional experiment focusing on the first-round fine-tuned model—trained solely with human-annotated data—would be valuable. This would clarify the necessity and impact of the iterative training approach.**
>
> A3: Thank you for your suggestions. We tested the first-round fine-tuned model and also conducted a first-round test on Diffusion-DPO using the same dataset. The results are as follows:
>
> | Model                      | Color$\uparrow$ | Shape$\uparrow$ | Texture$\uparrow$ | Spatial$\uparrow$ | Non-Spatial$\uparrow$ | Complex$\uparrow$ |
> | -------------------------- | --------------- | --------------- | ----------------- | ----------------- | --------------------- | ----------------- |
> | SDXL                       | 0.6369          | 0.5408          | 0.5637            | 0.2032            | 0.3110                | 0.4091            |
> | Diffusion-DPO(first round) | 0.6822          | 0.5691          | 0.6224            | 0.2216            | 0.3291                | 0.4263            |
> | IterComp(first round)      | 0.7239          | 0.6083          | 0.6940            | 0.2692            | 0.3308                | 0.4572            |
> | IterComp                   | **0.7982**      | **0.6217**      | **0.7683**        | **0.3196**        | **0.3371**            | **0.4873**        |
>
> From the results, it is evident that IterComp effectively leverages the model gallery to collect composition-aware model preferences, enabling it to learn diverse compositional dimensions during the first round of training. Notably, attributes such as color, texture, and spatial relationships exhibit significant improvement. Compared to Diffusion-DPO, the lack of explicitly trained reward models leads to mixed training on image pairs with multiple preferences, which weakens the model's ability to learn specific compositional preferences. For instance, its improvements in spatial relationships are notably limited, highlighting that the DPO approach is not well-suited for complex tasks like compositional generation.  Moreover, IterComp’s iterative optimization approach delivers substantial advantages across all metrics, further underscoring the effectiveness of iterative feedback learning in addressing complex compositional tasks.
>
>
>
> **Q4: Based on my experience, RLHF in diffusion models can often be unstable. I’m curious whether your method consistently produces stable results or if there’s a risk of occasionally poorer outcomes. I’m concerned that the iterative training process might lead to overfitting on biases present in the reward models, potentially reducing overall robustness.**
>
> A4: This is an excellent question. Regarding the stability of IterComp, we conducted experiments **in the appendix A.3 of the updated paper**. We selected five baseline methods—SD1.5, SDXL, InstanceDiffusion, Diffusion-DPO, and FLUX—along with two evaluation metrics: Complex and CLIP-Score. Using the same 50 random seeds, we calculated the mean and variance of each model under these two metrics. To visualize stability, we used the variance of each algorithm as the radius and scaled it uniformly by a factor of 10^4 for clarity.
>
> In terms of compositional stability, **as shown in Figure 8 (a)**, we found that IterComp not only achieved the best overall performance but also exhibited the highest stability. This is attributed to the iterative optimization process, which refines the model by analyzing and improving its output samples during each iteration. The iterative training approach allows the model to perform feedback learning on its own generated outputs, rather than relying solely on external datasets. This significantly enhances the model’s stability. When evaluating stability in terms of realism and generation quality, **as shown in Figure 8 (b)**, our method also achieved the best stability. These findings demonstrate that the iterative training approach not only significantly improves model performance but also greatly enhances its stability.
>
> To address the issue of overfitting, we adopted a targeted approach for training the reward model during each iteration. Specifically, we only fine-tuned the reward model using image pairs that include newly generated samples. For example, if the newly generated image is $x_0$, the reward model is fine-tuned on datasets such as $(x_0, x_{SDXL}), (x_0, x_{FLUX})$, while excluding repeated training on pairs like $(x_{SDXL}, x_{FLUX})$. This approach effectively prevents overfitting during multiple iterations while enabling the reward model to better learn composition-aware model preferences. As a result, the reward model can consistently refine and improve the optimized model in each iteration, achieving both self-correction and self-improvement.

---

> > ### Comment · Reviewer_bbqi · 2024-11-22
> >
> > Thank you for your detailed explanation and comprehensive experiments. I think that IterComp is an excellent work that fully leverages the capabilities of existing models through iterative training, enabling the development of a better-performing model with shorter inference time. I truly appreciate the method you proposed and the solid experimental results, and have raised my score to 8.
> >
> > Once again, thank you for your contributions to the research on generative models and for this insightful discussion.

---

> ### Author Response · Authors · 2024-11-22
> **Thank you for your support**
>
> Dear Reviewer bbqi，
>
> Thank you for raising score! We greatly appreciate your recognition of our work and the valuable feedback you provided. We will continue to optimize this method and strive to make more contributions to the field.
>
> Warm Regards,
>
> The Authors

---

### Official Review · Reviewer_HHze · 2024-11-03

**Soundness:** 3
**Presentation:** 3
**Contribution:** 3
**Rating:** 6
**Confidence:** 4

**Summary:**

This paper propose a framework that aggregates composition-aware model preferences from multiple models and employs an iterative feedback learning approach to enhance T2I compositionality and general performance. The qualitative and quantitative results show their SOTA compositional generation capabilities compared to previous works.

**Strengths:**

1. This paper presents a novel framework combining preferences from multiple diffusion models to enhance compositional text-to-image generation and address the relationship understanding in diffusion models.
2. The qualitative/quantitative results show comparable improvements in compositionality.
3. A composition-aware dataset is collected which provide diverse preferences that inform the reward models. (will it be released in the future?)

**Weaknesses:**

1. There is limited discussion of the computation resources required to manage multiple reward models, which may affect the scalability in large-scale applications. Although the authors claim that their model has fast inference speed, the cost of model training and data collection is not clear. This makes me feel less likely than DPO to be widely used in practice.
2. The user study only demonstrates the user preferences lacking the deep analysis of attribute binding and object relationship, which are critical to model performance. 16 samples is also too small to evaluate such a complex task.

**Questions:**

1. As mentioned in W.1, how long does the training loop take (including the iterative feedback learning)?
2. Could I use this method to improve a specific concept generation (e.g., a human-object interaction)? How much time does it take from collecting synthetic data to finalizing the model training?

---

> ### Author Response · Authors · 2024-11-21
> **Response to Reviewer HHze (Part 1/2)**
>
> *We sincerely thank you for your time and efforts in reviewing our paper, and your valuable feekback. We are glad to see that the proposed method is novel, the experimental results are promising, and we contribute a high-quality composition-aware dataset. Please see below for our responses to your comments.*
>
> **Q1: Will the composition-aware dataset be released in the future?**
>
> A1: We will fully open-source the composition-aware model preference dataset and the three pretrained reward models upon acceptance of the paper.
>
> **Q2: There is limited discussion of the computation resources required to manage multiple reward models, which may affect the scalability in large-scale applications. Although the authors claim that their model has fast inference speed, the cost of model training and data collection is not clear. This makes me feel less likely than DPO to be widely used in practice.**
>
> A2: This is an excellent question. The focus of our method lies in the concepts of the **model gallery** and the **iterative feedback learning** paradigm, both of which are highly generalizable. IterComp can be applied to a wide range of tasks, with the number and design of reward models remaining flexible and customizable, as they are not the core focus of our approach.  Given the inherent challenges of compositional generation, we employed three reward functions to enhance model performance, but this component is adjustable.
>
> While our method can be tailored to tackle complex tasks, whereas DPO is limited to handling simpler tasks. Without explicit reward models, DPO is unable to enable the base model to effectively learn from multiple preferences. Furthermore, due to the lack of reward models in DPO, it is not possible to automatically rank the generated images from the optimized base model or the newly added models, making it incompatible with the iterative feedback learning paradigm. We applied DPO to SDXL using the same dataset of IterComp (i.e., 52,500 image pairs), and the results on T2I-CompBench are as follows:
>
> | Model         | Color$\uparrow$ | Shape$\uparrow$ | Texture$\uparrow$ | Spatial$\uparrow$ | Non-Spatial$\uparrow$ | Complex$\uparrow$ |
> | ------------- | --------------- | --------------- | ----------------- | ----------------- | --------------------- | ----------------- |
> | SDXL          | 0.6369          | 0.5408          | 0.5637            | 0.2032            | 0.3110                | 0.4091            |
> | Diffusion-DPO | 0.6822          | 0.5691          | 0.6224            | 0.2216            | 0.3291                | 0.4263            |
> | IterComp      | **0.7982**      | **0.6217**      | **0.7683**        | **0.3196**        | **0.3371**            | **0.4873**        |
>
> As shown in the table, Diffusion-DPO (i.e., the model optimized using DPO on the same training set as IterComp) shows a significant gap compared to IterComp, with limited improvements in aspects such as shape and spatial relationships. This is because DPO's training method is more suited for simpler generation tasks, and the lack of an explicit reward model restricts the model's ability to learn composition-aware model preferences. Additionally, it cannot improve the model's capabilities through iterative training.
>
> **Q3: The user study only demonstrates the user preferences lacking the deep analysis of attribute binding and object relationship, which are critical to model performance. 16 samples is also too small to evaluate such a complex task.**
>
> A3: Thank you for your suggestion. We conducted a larger and more comprehensive user study, **with the results included in the appendix A.4 of the updated paper**. The study involved 41 randomly selected participants from diverse backgrounds. We compared IterComp with five other methods across four aspects: attribute binding, spatial relationships, non-spatial relationships and overall performance. Each comparison involved 25 prompts, culminating in a final survey of 125 prompts and generating 20,500 votes. From the win rate distribution of IterComp shown in the figure, it is evident that IterComp demonstrates significant advantages across all three aspects of compositional generation.
>
> Specifically, compared to the layout-based model InstanceDiffusion, IterComp shows an absolute advantage in attribute binding. For text-based models such as SDXL and FLUX, IterComp leads significantly in spatial relationships. This highlights that the model gallery design effectively collects composition-aware model preferences and enhances performance across different compositional aspects through iterative feedback learning.

---

> ### Author Response · Authors · 2024-11-21
> **Response to Reviewer HHze (Part 2/2)**
>
> **Q4: As mentioned in W.1, how long does the training loop take (including the iterative feedback learning)?**
>
> A4: Here is an approximate listing of the training time for each stage (all our experiments were conducted on 4*NVIDIA A100-80G GPUs):
>
> | Phrase                                   | Training Time |
> | ---------------------------------------- | ------------- |
> | Train reward models (iteration1)         | 31min         |
> | Train base diffusion models (iteration1) | 3h 37min      |
> | Train reward models (iteration2)         | 14min         |
> | Train base diffusion models (iteration2) | 1h 31min      |
> | Train reward models (iteration3)         | 17min         |
> | Train base diffusion models (iteration3) | 1h 31min      |
> | Total                                    | ~7h 41min     |
>
> To prevent overfitting and to improve the self-correction of the reward model, from the second iteration, we only fine-tuned the reward model using image pairs that include newly generated samples. For example, if the newly generated image is $x_0$, the reward model is fine-tuned on datasets such as $(x_0, x_{SDXL}), (x_0, x_{FLUX})$, while excluding repeated training on pairs like $(x_{SDXL}, x_{FLUX})$. This approach effectively prevents overfitting during multiple iterations while enabling the reward model to better learn composition-aware model preferences. Moreover, from the second iteration, the optimization of base diffusion models reduces the number of epochs by half, from 2 to 1. As a result, with the continuous refinement of iterative feedback learning, the training time of the model decreases progressively. Ultimately, the total training time is approximately 7 hours and 41 minutes.
>
>
> **Q5: Could I use this method to improve a specific concept generation (e.g., a human-object interaction)? How much time does it take from collecting synthetic data to finalizing the model training?**
>
> A5: Our IterComp is a versatile optimization framework that can be effectively tailored to enhance specific concept generations. For instance, in the context of human-object interactions, we could employ powerful MLLMs or domain-specific models to generate multiple rewards from various perspectives, including the accuracy of human joint positions, object translation and rotation, and human-object contact. Once these reward models are in place, we can leverage an IterComp-like pipeline to iteratively refine the base generation model. As demonstrated in our research, the optimization process typically converges within three iterations, making it highly efficient in practice. Moreover, the amount of synthetic data required for optimization is minimal, approximately 1/1000 of the data needed for training the base model. Consequently, the overall time required is significantly less than that for training the original generation model.

---

### Official Review · Reviewer_52Cf · 2024-11-04

**Soundness:** 2
**Presentation:** 3
**Contribution:** 2
**Rating:** 6
**Confidence:** 3

**Summary:**

This paper introduces IterComp, a novel framework that enhances compositional text-to-image generation by aggregating preferences from multiple diffusion models through iterative feedback learning. The approach demonstrates superior performance in both compositional accuracy and image quality, while maintaining efficient inference speed. The main strength lies in its ability to combine different models' advantages without adding computational overhead, though the long-term stability of the iterative learning process could be further explored.

**Strengths:**

Novel and practical approach: The paper presents a simple yet effective way to combine multiple models' strengths for compositional generation without increasing computational complexity.

Strong empirical results: The method shows clear improvements over existing approaches, with comprehensive evaluations on both compositional accuracy and image quality metrics.

Well-structured technical contribution: The paper provides clear theoretical analysis with detailed proofs, and the iterative feedback learning framework is well-designed and easy to implement.

**Weaknesses:**

The paper mentioned that RPG is challenging to achieve precise generation, but Tab2 did not compare with RPG, and I checked that RPG's performance on T2I-Compbench is better than that of the paper.

It is necessary to test the results of FLUX-dev directly on the t2i compbench to see how much improvement the method proposed in this paper has. I currently suspect that the improvement may not be very significant.

**Questions:**

See weakness

---

> ### Author Response · Authors · 2024-11-21
> **Response to Reviewer 52Cf (Part 1/2)**
>
> *We sincerely thank you for your time and efforts in reviewing our paper, and your valuable feekback. We are glad to see that the proposed method is novel and practical, the experimental results are strong empirical, the theoretical analysis is detailed and our method is well-designed and easy to follow. Please see below for our responses to your comments.*
>
> **Q1: The paper mentioned that RPG is challenging to achieve precise generation, but Tab2 did not compare with RPG, and I checked that RPG's performance on T2I-Compbench is better than that of the paper.**
>
> A1: Thank you for your comment! IterComp is a general framework for text-to-image generation or diffusion alignment, rather than an improvement for a specific model. It is worth noting that RPG uses MLLM to handle generation tasks with complex prompts.
> Such LLM-Enhanced diffusion models naturally perform better than pure diffusion models (e.g., SDXL and our IterComp) on complex generation. To make fair comparison, we replace the diffusion backbone of RPG with our IterComp and provide its metrics on T2I-CompBench:
>
> | Model        | Color$\uparrow$ | Shape$\uparrow$ | Texture$\uparrow$ | Spatial$\uparrow$ | Non-Spatial$\uparrow$ | Complex$\uparrow$ | Inference Time$\downarrow$ |
> | ------------ | --------------- | --------------- | ----------------- | ----------------- | --------------------- | ----------------- | -------------------------- |
> | IterComp     | 0.7982          | 0.6217          | 0.7683            | 0.3196            | 0.3371                | 0.4873            | **5.63 s/Img**             |
> | RPG          | 0.8335          | 0.6801          | 0.8129            | 0.4547            | 0.3462                | 0.5408            | 15.57 s/Img                |
> | RPG+IterComp | **0.8668**      | **0.7216**      | **0.8201**        | **0.4874**        | **0.3498**            | **0.5661**        | 15.57 s/Img                |
>
> The table shows that when the RPG backbone is replaced with IterComp, the model significantly outperforms across all six metrics on the T2I-CompBench. This highlights IterComp's superiority in compositional generation. It's important to note that IterComp is a simple SDXL-like model that doesn't require complex computations during inference. As a result, under the same conditions such as prompts and inference steps, IterComp is nearly **three times faster than RPG**.
>
> In addition, to more comprehensively evaluate the capabilities of IterComp and RPG in compositional generation, we employed two additional, up-to-date benchmarks for testing:
>
> DPG-Bench:
>
> | Model        | Global    | Entity    | Attribute | Relation  | Other     | Average   |
> | ------------ | --------- | --------- | --------- | --------- | --------- | --------- |
> | IterComp     | 89.91     | 88.64     | 86.73     | 84.77     | 89.74     | 81.17     |
> | RPG          | 91.01     | 87.39     | 84.53     | 87.92     | 89.84     | 81.28     |
> | RPG+IterComp | **92.74** | **91.33** | **89.10** | **92.38** | **90.13** | **84.72** |
>
> GenEval:
>
> | Model        | Single Obj. | Two Obj. | Counting | Colors   | Position | Color Attri. | Overall   |
> | ------------ | ----------- | -------- | -------- | -------- | -------- | ------------ | --------- |
> | IterComp     | 0.97        | 0.85     | 0.63     | 0.86     | 0.33     | 0.41         | 0.675     |
> | RPG          | 0.97        | 0.86     | 0.66     | 0.79     | 0.30     | 0.38         | 0.660     |
> | RPG+IterComp | **0.99**    | **0.90** | **0.72** | **0.90** | **0.35** | **0.48**     | **0.723** |
>
> As demonstrated in the two benchmarks above, IterComp outperforms RPG in metrics like attributes and colors. This is due to our training of a specific reward model for attribute binding, which iteratively enhances IterComp over multiple iterations. Leveraging the strong planning and reasoning capabilities of LLMs, RPG excels in areas such as relations, counting, and positioning. When IterComp is used as the backbone for RPG, the model exhibits remarkable performance across all aspects. We have **included this experiment in the appendix A.5 of the updated manuscript**. Thank you again for your feedback.

---

> > ### Comment · Reviewer_52Cf · 2024-11-25
> > **Review response**
> >
> > Hi authors,
> >
> > The rebuttal solves my concerns. I decide to increase my score to 6.

---

> > > ### Author Response · Authors · 2024-11-25
> > > **Thank you for your support**
> > >
> > > Dear Reviewer 52Cf，
> > >
> > > Thank you for raising score! We sincerely appreciate your valuable comments and your precious time in reviewing our paper!
> > >
> > > Warm Regards,
> > >
> > > The Authors

---

> ### Author Response · Authors · 2024-11-21
> **Response to Reviewer 52Cf (Part 2/2)**
>
> **Q2: It is necessary to test the results of FLUX-dev directly on the t2i compbench to see how much improvement the method proposed in this paper has. I currently suspect that the improvement may not be very significant.**
>
> A2: Thank you for your suggestion! We have provided the T2I-CompBench results for FLUX and SD3 from the model gallery as follows:
>
> | Model          | Color$\uparrow$ | Shape$\uparrow$ | Texture$\uparrow$ | Spatial$\uparrow$ | Non-Spatial$\uparrow$ | Complex$\uparrow$ | Inference Time$\downarrow$ |
> | -------------- | --------------- | --------------- | ----------------- | ----------------- | --------------------- | ----------------- | -------------------------- |
> | SDXL           | 0.6369          | 0.5408          | 0.5637            | 0.2032            | 0.3110                | 0.4091            | **5.63 s/Img**             |
> | SD3-medium     | 0.7442          | 0.6053          | 0.7055            | 0.2419            | 0.3294                | 0.4548            | 13.44 s/Img                |
> | FLUX-dev       | 0.7881          | 0.6529          | 0.7472            | 0.2606            | 0.3396                | 0.4731            | 23.02 s/Img                |
> | IterComp(SDXL) | **0.7982**      | 0.6217          | **0.7683**        | **0.3196**        | 0.3371                | **0.4873**        | **5.63 s/Img**             |
> | IterComp(SD3)  | **0.8532**      | **0.6922**      | **0.8493**        | **0.4074**        | **0.3482**            | **0.5419**        | 13.44 s/Img                |
>
>
>
> As shown in the table, IterComp demonstrates a significant advantage over SD3 and FLUX in terms of spatial-relationship, and offers faster generation speeds. This is because SD3 and FLUX have limited spatial-awareness, whereas our approach iteratively enhances the model’s spatial-awareness through a reward function focused on spatial relationships. It is worth noting that the original IterComp in our paper is optimized based on SDXL, and our main focus is to illustrate how this new optimization framework improves existing diffusion models.
> Thus it is unfair to make a direct comparison with models like SD3 and FLUX. Besides, according to this table, our IterComp can significantly improve the performance of SD3 and outperform previous models (including FLUX) from all aspects of compositional generation, demonstrating the effectiveness of our IterComp framework.

---

### Official Review · Reviewer_Vtxw · 2024-11-04

**Soundness:** 3
**Presentation:** 3
**Contribution:** 3
**Rating:** 6
**Confidence:** 3

**Summary:**

This paper proposes IterComp, an iterative composition-aware reward-controlled framework. It introduces a model gallery and constructs a high-quality composition-aware model preference dataset. Utilizing a new iterative feedback learning framework, IterComp progressively enhances both the reward models and the base diffusion model.

**Strengths:**

1. This work claims to be the first work to introduce a reward-controlled framework in the concept composition generation, which is somehow novel in this field.
2. This work is presented well with complete theoretical details and proof.
3. The quantitative experimental results show better performance compared to SOTAs.

**Weaknesses:**

1. The qualitative comparison in Fig. 4 is confused. There seems to be a marginal improvement in Line 3 and Line 4. The authors should make the difference between the qualitative examples clear to be recognized.
2. The qualitative examples are not enough to evaluate the model performance since the cases in the paper are somehow complex. The authors can provide more examples for a single case. Also, there is a need to evaluate the stability of the proposed model.
3. The comparison with InstanceDiffusion is confusing. As a layout-guided method, InstanceDiffusion needs detailed layout inputs. It is not fair to provide only one box if the case includes two or more instances, as indicated in the third line of Fig. 4. As the authors attempt to compare with layout-to-image methods, a SOTA method named MIGC [1] is also not included.

[1] Zhou, Dewei, et al. "Migc: Multi-instance generation controller for text-to-image synthesis." Proceedings of the IEEE/CVF Conference on Computer Vision and Pattern Recognition. 2024.

**Questions:**

Please see the weakness.

---

> ### Author Response · Authors · 2024-11-21
> **Response to Reviewer Vtxw (Part 1/2)**
>
> *We sincerely thank you for your time and efforts in reviewing our paper, and your valuable feekback. We are glad to see that the proposed method is novel, the theoretical details and proof are complete, and the experimental results are promising. Please see below for our responses to your comments.*
>
> **Q1: The qualitative comparison in Fig. 4 is confused. There seems to be a marginal improvement in Line 3 and Line 4. The authors should make the difference between the qualitative examples clear to be recognized.**
>
> A1: We apologize for any inconvenience caused. The examples in the paper were not deliberately selected, and to demonstrate the superiority of our method, we conducted a large number of additional experiments **in the appendix A.7 of the updated manuscript**. Furthermore, to better showcase the advantages of IterComp over other algorithms, we have added evaluations against FLUX and SD3 on T2I-CompBench:
>
> | Model      | Color$\uparrow$ | Shape$\uparrow$ | Texture$\uparrow$ | Spatial$\uparrow$ | Non-Spatial$\uparrow$ | Complex$\uparrow$ | Inference Time$\downarrow$ |
> | ---------- | --------------- | --------------- | ----------------- | ----------------- | --------------------- | ----------------- | -------------------------- |
> | SDXL       | 0.6369          | 0.5408          | 0.5637            | 0.2032            | 0.3110                | 0.4091            | **5.63 s/Img**             |
> | SD3-medium | 0.7442          | 0.6053          | 0.7055            | 0.2419            | 0.3294                | 0.4548            | 13.44 s/Img                |
> | FLUX-dev   | 0.7881          | **0.6529**      | 0.7472            | 0.2606            | **0.3396**            | 0.4731            | 23.02 s/Img                |
> | IterComp   | **0.7982**      | 0.6217          | **0.7683**        | **0.3196**        | 0.3371                | **0.4873**        | **5.63 s/Img**             |
>
> As shown in the table, IterComp demonstrates strong superiority in compositional generation, outperforming SD3. Despite FLUX’s massive 12B parameters, **IterComp achieves similar or even better performance with only 1/5 of the parameters and generates images nearly 4 times faster**, with some metrics comparative to FLUX, while the majority demonstrate stronger results. IterComp is a highly generalizable text-to-image or diffusion alignment framework, and in the future, we plan to adapt IterComp to FLUX, which will further showcase even greater generative capabilities.
>
> **Q2: The qualitative examples are not enough to evaluate the model performance since the cases in the paper are somehow complex. The authors can provide more examples for a single case.**
>
> A2: We thank the reviewer for helping us improve our paper. We have added more experiments **in the appendix A.7 of the updated manuscript**, including comparisons with additional simple examples. Please refer to the updated manuscript for details.
>
> **Q3: There is a need to evaluate the stability of the proposed model.**
>
> A3: Thank you for your suggestions! We conducted experiments on the method's stability, **detailed in the appendix A.3 of the updated manuscript**. We selected five methods for comparison: SD1.5, SDXL, InstanceDiffusion, Diffusion-DPO, and FLUX, along with two evaluation metrics: *complex* and *CLIP-score*. Using the same 50 seeds, we calculated the mean and variance of the models’ performance for these metrics. To facilitate visualization, we used the variance of each method as the radius and scaled it uniformly by a common factor (10^4) for stability analysis.
> Regarding the stability of compositionality, **as shown in Figure 8(a)**, we found that IterComp not only achieved the best overall performance but also demonstrated superior stability. This can be attributed to the iterative feedback learning paradigm enable the model to **analyze and refine its output at each optimization step, effectively self-correcting and self-improving**. The iterative training approach enables the model to perform feedback training based on its own generated samples rather than solely relying on external data, this enables the model to steadily improve over multiple iterations based on its own foundation, which significantly enhances model stability.
>
> For the stability of realism or generation quality, **as shown in Figure 8(b)**, our method also exhibited the highest stability. Therefore, the iterative training approach not only improves the model's performance but also substantially enhances its stability across different dimensions.

---

> ### Author Response · Authors · 2024-11-21
> **Response to Reviewer Vtxw (Part 2/2)**
>
> **Q4: The comparison with InstanceDiffusion is confusing. As a layout-guided method, InstanceDiffusion needs detailed layout inputs. It is not fair to provide only one box if the case includes two or more instances, as indicated in the third line of Fig. 4. As the authors attempt to compare with layout-to-image methods, a SOTA method named MIGC [1] is also not included.**
>
> A4: We are grateful for your feedback and apologize for any potential confusion caused. For the layout-to-image method, we used GPT-4o to infer the corresponding layout for each example in the paper. Since layout-based models are not the focus of our work, we chose not to include these layouts in the visualizations. However, it's important to note that **all examples of layout-based models were generated with precise layouts**, as mentioned in line 408 of the original paper. We apologize for any misunderstandings and have **clarified this in the updated manuscript**.
>
> Additionally, in the **appendix  A.6 of the updated manuscript**, we provide additional experiments between IterComp, InstanceDiffusion, and MIGC. These examples clearly show that while MIGC and InstanceDiffusion can accurately generate objects in the specified positions of the layout, there is a notable gap in generation quality compared to IterComp, such as aesthetics and details. Moreover, the images generated by these two methods often appear visually unrealistic, with significant flaws such as incomplete violins or mismatches between bicycle and its basket. This highlights the clear superiority of our method. Additionally, we have included an evaluation of MIGC on T2I-CompBench:
>
> | Model             | Color$\uparrow$ | Shape$\uparrow$ | Texture$\uparrow$ | Spatial$\uparrow$ | Non-Spatial$\uparrow$ | Complex$\uparrow$ | Inference Time$\downarrow$ |
> | ----------------- | --------------- | --------------- | ----------------- | ----------------- | --------------------- | ----------------- | -------------------------- |
> | InstanceDiffusion | 0.5433          | 0.4472          | 0.5293            | 0.2791            | 0.2947                | 0.3602            | 9.88 s/Img                 |
> | MIGC              | 0.5914          | 0.4503          | 0.5162            | 0.2811            | 0.2908                | 0.3729            | 11.47 s/Img                |
> | IterComp          | **0.7982**      | **0.6217**      | **0.7683**        | **0.3196**        | **0.3371**            | **0.4873**        | **5.63 s/Img**             |
>
> As shown in the table, IterComp outperforms across all metrics and achieves nearly double the generation speed. Thank you once again for your valuable suggestions!

---

### Author Response · Authors · 2024-11-21
**Global Response**

We sincerely thank all the reviewers for their thorough reviews and valuable feedback. We are glad to hear that our proposed framework is novel and practical (reviewer Vtxw, 52Cf, HHze and ocHL), the theoretical details and proof is clear and complete (reviewer Vtxw, 52Cf and bbqi), and the performance improvements demonstrated in experiments are significant and promising (all reviewers).

We summarize our responses to the reviewers' comments as follows:

- We additionally provide more examples and conduct more experiments to show the significant improvement of our IterComp and **updated our manuscript in Appendix. A.5, A.6, and A.7**. (Reviewer Vtxw, 52Cf, bbqi and ocHL)
- We additionally conduct experiments on model stability and **updated our manuscript in Appendix. A3** (Reviewer Vtxw and bbqi)
- We additionally conduct an analysis of the training time and application of IterComp (Reviewer HHze, bbqi and ocHL)

We reply to each reviewer's questions in detail below their reviews. Please kindly check out them. Thank you and please feel free to ask any further questions.

---

### Meta-Review · Area_Chair_5gWS · 2024-12-09

**Metareview:**

All reviewers agree to accept the paper. Reviewers appreciate the novel framework and significant performance improvement. Please be sure to address the reviewers' comments in the final version.

**Additional Comments On Reviewer Discussion:**

All reviewers agree to accept the paper.

---

### Decision · Program_Chairs · 2025-01-22

Accept (Poster)